# Bounds to electron spin qubit variability for scalable CMOS architectures

Jesús D. Cifuentes [1] ✉, Tuomo Tanttu [1,2], Will Gilbert [1,2], Jonathan Y. Huang [1], Ensar Vahapoglu [1,2], Ross C. C. Leon[1], Santiago Serrano [1], Dennis Otter[1], Daniel Dunmore[1], Philip Y. Mai[1], Frédéric Schlattner[1,3], MengKe Feng [1], Kohei Itoh [4], Nikolay Abrosimov [5], Hans-Joachim Pohl[6], Michael Thewalt[7], Arne Laucht [1,2], Chih Hwan Yang [1,2], Christopher C. Escott[1,2], Wee Han Lim[1,2], Fay E. Hudson [1,2], Rajib Rahman [8], Andrew S. Dzurak [1,2] & Andre Saraiva [1,2] ✉

Spins of electrons in silicon MOS quantum dots combine exquisite quantum properties and scalable fabrication. In the age of quantum technology, however, the metrics that crowned Si/SiO$_2$ as the microelectronics standard need to be reassessed with respect to their impact upon qubit performance. We chart spin qubit variability due to the unavoidable atomic-scale roughness of the Si/SiO$_2$ interface, compiling experiments across 12 devices, and develop theoretical tools to analyse these results. Atomistic tight binding and path integral Monte Carlo methods are adapted to describe fluctuations in devices with millions of atoms by directly analysing their wavefunctions and electron paths instead of their energy spectra. We correlate the effect of roughness with the variability in qubit position, deformation, valley splitting, valley phase, spin-orbit coupling and exchange coupling. These variabilities are found to be bounded, and they lie within the tolerances for scalable architectures for quantum computing as long as robust control methods are incorporated.

The interface between silicon and its oxide permeates most of the technology that enabled the digital era, and as such it is one of the most studied materials in human history. The practical process engineering advantages of silicon dioxide contrast with the complex chemistry of this material, and the decades of research that has underpinned the development of the CMOS industry. As we approach a new era of quantum technology, this know-how is largely considered a major advantage for technologies such as silicon-based spin qubits[1]. In particular, the similarity between quantum dots defined by gate electrodes on top of a silicon/silicon dioxide interface and the MOSFET transistors in materials, design, and fabrication enables the integration of manufacturing techniques exclusive to semiconductor foundries onto the scaling of quantum processors[2].

Here, we specifically treat the case of such qubits formed in quantum dots at the Si/SiO$_2$ interface, which are compatible with the high-yield integration of on-chip electronic components, and refer to these as CMOS qubits. We note, however, that other forms of silicon-based quantum dots can be manufactured, for instance, by leveraging a Si/SiGe quantum well[3]. These materials present their own complex challenges and advantages and impose significantly different architectural choices compared to the Si/SiO$_2$ interface, and hence are left out of this investigation.

[1]School of Electrical Engineering and Telecommunications, University of New South Wales, NSW 2052 Sydney, NSW, Australia. [2]Diraq, Sydney, NSW, Australia. [3]Solid State Physics Laboratory, Department of Physics, ETH Zurich, Zurich 8093, Switzerland. [4]School of Fundamental Science and Technology, Keio University, Yokohama, Japan. [5]Leibniz-Institut für Kristallzüchtung, 12489 Berlin, Germany. [6]VITCON Projectconsult GmbH, 07745 Jena, Germany. [7]Department of Physics, Simon Fraser University, V5A 1S6 Burnaby, BC, Canada. [8]School of Physics, University of New South Wales, Sydney, NSW 2052, Australia. ✉e-mail: j.cifuentes_pardo@unsw.edu.au; a.saraiva@unsw.edu.au

Despite a relatively late start[4], the performance of silicon qubits has led to fidelity levels comparable with more well-established quantum technologies like superconducting or ion-trap qubits[5–7]. The recent demonstration of repeatable high-fidelity two-qubit operations across three nominally identical CMOS devices[8] signals the beginning of an age focused on extensive repeatability of high performance to achieve scaled architectures. However, despite the improvements in gate uniformity demonstrated by the integration of foundry-level manufacturing techniques[9], new concerns are stirred by the fragility of spins to effects that are ignored in classical transistor technology, such as variations in spin–orbit coupling (impacting one-qubit frequencies), exchange interaction (two-qubit couplings) and valley splitting (nearest excitation energy).

It is only by understanding this variability quantitatively that it is possible to develop a sensible, scalable quantum processor architecture. Enabling the breakthrough applications of quantum computation requires millions of qubits to perform error correction[10,11], and it is infeasible to address all of these qubits with pulses catering to their particular parameters with wires individually running from the room temperature controllers. Instead, this variability must be embraced and corrected through the combination of on-chip electronics operating at cryogenic temperatures[2,12], and robust quantum control pulses that will be shared among several qubits[13,14]. Designing control cells that can offset qubit properties across a large range and with sufficient accuracy is only possible if the electric tunability compensates for the range of variations. As we will discuss in this paper, this interplay between variability and tunability differs between qubit parameters (one-qubit frequencies and two-qubit couplings, etc.), so a different control strategy must be adopted in each scenario.

Qubit variability is caused by the same factors that affect conventional CMOS technology, such as strain, fabrication defects, accidental introduction of charged impurities in the oxide, interface roughness, and so on[15]. Industrial foundries focus most of their efforts on addressing these issues[9,16,17], with a central role played by the choice of materials for substrate, dielectrics, gate metals, etc.[18].

In the centre of the discussion is the dielectric interface where the electrons are confined[19,20]. High-fidelity silicon qubits have been measured in Si/SiGe and Si/SiO$_2$ heterostructures[3,5–8]. In the case of Si/SiGe heterostructures, a thin layer of uniaxially strained silicon binds the electron due to the conduction band shift caused by the strain when compared to the relaxed Si$_x$Ge$_{1−x}$ alloy. The Si/SiGe interface provides, in general, reduced levels of interfacial disorder compared to that of Si/SiO$_2$[19]. Potential shortcomings of Si/SiGe technology are the reduced gate control when compared to MOS devices and the limited tolerance of the material stack to high-temperature annealing processes, commonly adopted in the CMOS industry. More information on this material and its comparison to oxides can be found in ref. 18. In the context of spin qubit variability, comparing an oxide interface to Si/SiGe is hard because the nature of the disorder in the two materials is different—alloy disorder and miscut angles in SiGe, compared to amorphous oxidation in SiMOS. Moreover, the dominant effect of spin–orbit coupling being studied here is masked by the presence of micromagnets, which are commonly adopted in SiGe qubit architectures[3].

In this paper, we focus on qubits at the Si/SiO$_2$ interface. SiO$_2$ does not have a regular lattice structure when thermally grown on the silicon surface, so the interface is atomically rough. The higher levels of interfacial disorder when compared to Si/SiGe are attributed to this roughness and to the presence of fixed charge defects that can be either at the Si/SiO$_2$ interface, in the bulk of SiO$_2$ or at the metal-oxide interface[16,17,19,21]. Potential advantages are in the higher electrical tunability and compatibility with conventional CMOS technology, which benefits the integration with on-chip electronics[2].

The roughness of the Si/SiO$_2$ interface is one of the most critical sources of disorder for CMOS spin qubits[22–27]. A more recent paper indicated that a second source - charged impurities in the oxide - could dominate over interface roughness[28], at least in the case where a micromagnet is integrated to allow electron spin qubits to be driven electrically. Electric driving requires a large spin–orbit coupling, which exposes the spins to the impact of charge impurities. In this paper we focus on spin qubits driven magnetically[4,29], which do not have this requirement. These qubits can be controlled coherently without the inclusion of micromagnets, thus preserving the low spin–orbit coupling of electrons in silicon and protecting the spin from electric fluctuations. We will show in this paper, that under these conditions, the remaining spin–orbit variability is interface-induced, with charge impurities only affecting the qubit by shifting the quantum dot formation against the roughness profile of the interface.

Recent advances in CMOS quantum dot fabrication allowed for sufficient yield to create a number of small-scale quantum processing units and measure their variability. This work combines measurements of 12 qubits across 6 different CMOS quantum dot devices, transmission electron microscopy (TEM) images of cross-sectional cuts of 6 other quantum dot devices, and theoretical analysis of quantum properties of electrons in simulated quantum arrays of 49 quantum dots (Fig. 1a). Such a geometry allows us to study the impact of the self-affine scaling of the Si/SiO$_2$ roughness on qubit properties at different length-scales[30] (Fig. 1b). Starting from a detailed view to the device architecture, quantum dot formation, and materials interface down to the atomic scale, we predict the bounds to spin qubit variations. This prediction is compared with data from qubit devices, some of which have led to manuscripts and publications (see Table 1). All devices were fabricated with geometrically identical designs (see structure depicted in Fig. 1c) and differ only in material stack compositions and spin control methods (Table 1).

## Results

### Si/SiO$_2$ roughness

The Si/SiO$_2$ interface has an intrinsic fractal structure[30], that has not been considered in previous variability studies (refs. 26–28). Our decision to simulate a $7 \times 7$ dot array is motivated by understanding how this fractal scaling of the interface roughness impacts qubit properties at different length scales (see Fig. 1b). One-qubit properties, for instance, depend on the roughness obtained within the quantum dot diameter (-15 nm), while two-qubit properties are related to the interdot distance (30–60 nm). The roughness amplitude can differ significantly between these two scales due to this fractal structure, so a quantitative characterisation of the interface is necessary.

Our model of the Si/SiO$_2$ (Fig. 1b) is based on the roughness observed in TEM images (Fig. 1d, e)[30,31], allowing us to include more realistic features. By convolving these images with the expected face-centred cubic lattice of monocrystalline silicon, we can mathematically discern the interface and analyse the roughness at different scales[31], quantified through its power spectral density (PSD) as a function of the in-plane correlation length scale $\lambda$[30]. Our work incorporates a theoretical study of the realistic scale-dependent fractal structure of the roughness[30] and the development of theoretical tools capable of capturing this multiscale physics in a realistic device model (Fig. 1f, g).

Combining multiple TEM images, we obtained in Fig. 1h a consistent roughness pattern characteristic of a fractal scaling down to the silicon lattice parameter of the form $\mathrm{PSD}^{1D}(\lambda) \propto \left(\frac{2\pi}{\lambda}\right)^{-1-2H}$. We estimate a Hurst exponent of $H = 0.28$[32] (details in Supplementary Fig. 1 and Methods section). The root-mean-square roughness also scales up with the lateral region $\lambda$ as $\mathrm{RMS}(\lambda) \propto \left(\frac{2\pi}{\lambda}\right)^{-H}$. As shown in Fig. 1i this roughness pattern is consistent across all devices measured, and extends up to half a micrometre. We note that these levels of roughness are typical for industry-standard interfaces[17]. Our computer-

generated interface in Fig. 1b, f, g was also designed to mimic these features.

A direct conclusion from Fig. 1h, i is that the size of the dots (~15 nm for all devices studied) and the separation between dots (~50 nm) will have a large impact on the qubit exposure to surface roughness, and that the interface distortions within a quantum dot are smaller than those between neighbouring dots. The root-mean-square (RMS) is ~0.15 nm within a dot (~1 monolayer of the Si lattice), and almost twice for the double dot (RMS is 0.3 nm, ~2 monolayers of the Si lattice).

## Variability of quantum dot structure and excitation energy

The consistency of this roughness pattern across devices allows us to theoretically forecast its impact on qubit performance at scale. We simulate the spin qubit variability of quantum dots formed in different subsections of the computer-generated interface in Fig. 2a, b. Unless indicated differently, all quantum dots are simulated using the same electrostatic potential, which is based on COMSOL simulations of realistic digital models of our devices (see Fig. 1c–g and Methods section). The only difference between simulations is the location of the quantum dots in the surface profile. Such a model allows us to incor-

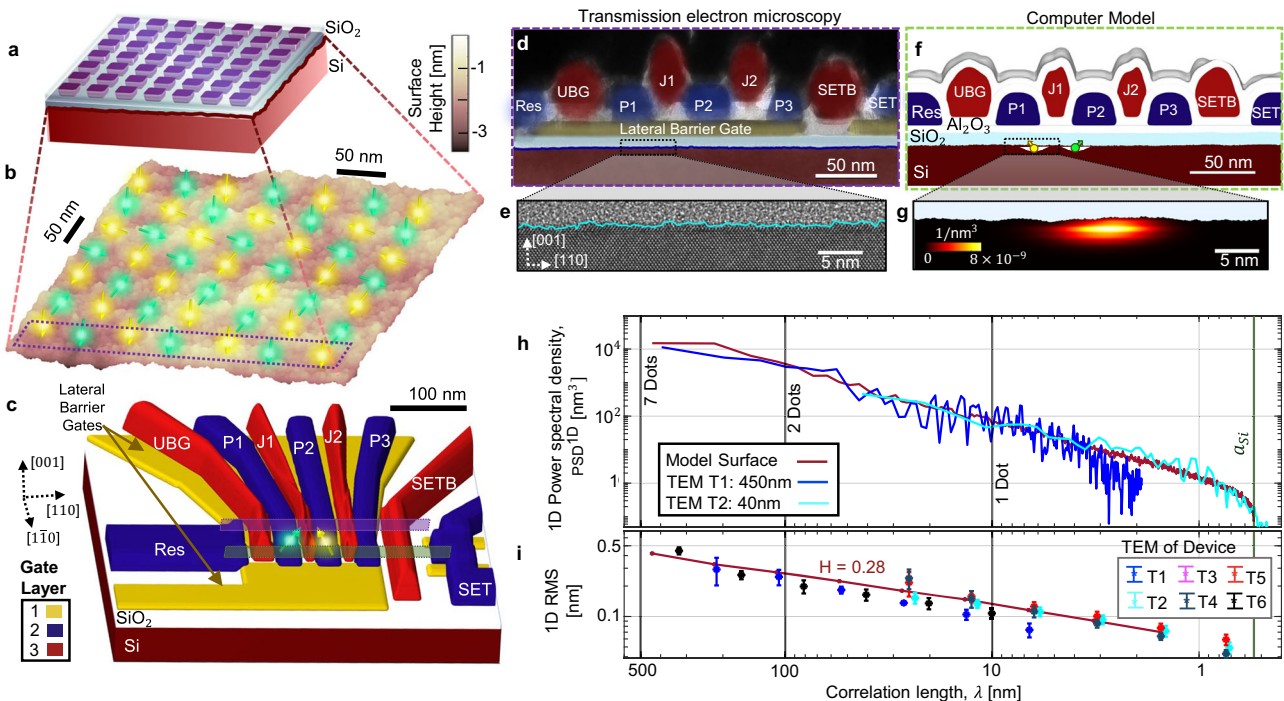

**Fig. 1 | Modelling CMOS spin qubits. a** Example scaled architecture of a 49 qubit device. **b** The quantum dots are formed below the computer-generated rough surface. **c** Model of the three-dot devices measured in this paper. The metallic gates are coloured by their order of deposition in different layers. **d–g** Comparison between device cross sections in TEM images and computer model. **d**, TEM image of device T1, showing a cross-section of the device located approximately at the position of the violet rectangular region in (**c**). **e** TEM of device T2 with a focus on the silicon oxide interface. We highlight in (**d**) a square region with the same size. **f** Cross-section of model at the green rectangular region in (**c**). **g** Atomistic simulation showing the electronic wavefunction of a quantum dot below rough Si-SiO₂. **h** Average power spectral density (PSD) of the Si/SiO₂ interface comparing the

interface from transmission electron microscopy (TEM) images of device T1 (blue) and T2 (**d**) (cyan), and the computer-generated surface in (**a**) (see also Supplementary Fig. 1 and Methods). The data were plotted as a function of $\lambda = \frac{2\pi}{q}$, where $q$ is the wave number. This allows us to compare $\lambda$ with the most relevant length scales, namely the silicon lattice parameter ($a_{Si} = 0.543$ nm), the dot diameter (10–15 nm), the double dot length (80–100 nm) and the lateral length of the simulation cell containing all 7 × 7 dots (500 nm). **i** Average RMS of segments of length $\lambda$ for the random surface generated numerically and for devices T1 to T6. Error bars indicate the standard error estimated from repeated measurements across multiple TEM images. (see Methods and Supplementary Table 1). Source data of figures **e**, **h**, **i** are provided in the Source Data file.

## Table 1 | List of devices used in qubit measurements

| Device | Dots | Configuration | Driving | Vector magnet | Gate material | Publications |
|--------|------|---------------|---------|---------------|---------------|--------------|
| **A** | P1, P2 | (1,1),(3,1),(1,3) | Antenna | Yes | Pd/Ti + ALD | 46 |
| **B** | P2, P3 | (3,1) | Antenna | No | Pd/Ti + ALD | - |
| **C** | P2, P3 | (3,1) | Antenna | No | Pd/Ti + ALD | 46 |
| **D** | P1, P2 | (1,3) | Dielectric Resonator | No | Pd/Ti + ALD | 29 |
| **E** | P1, P2 | (3,1) | Antenna | Yes | Al | 8,49,69 |
| **F** | P2 | 1 electron | Antenna | No | Al | 8,46 |

The devices are identical except for the differences indicated here. The data in device A is taken at three electronic configurations: (1,1) - for 1 electron under gate 1 and 1 electron under gate 2, (3,1) and (1,3). The double quantum dots are formed under the gates P1, P2 or P2, P3 depending on the device. In one of the devices the qubits were driven magnetically with a dielectric resonator instead of an antenna[29,70]. A vector magnet enabled rotations of the magnetic field for the measurements in two of the devices. The next column refers to the gate material. We use a combination of palladium and atomic layer deposition (ALD) alumina for some devices, and for others, we form the gates with aluminium and isolate them with thermally formed alumina. Differences between these situations are discussed in ref. 18. The last column shows the publications associated with each device. Device F is the only one with a single dot configuration instead of a double dot. We use this device only to provide additional data on the valley splitting.

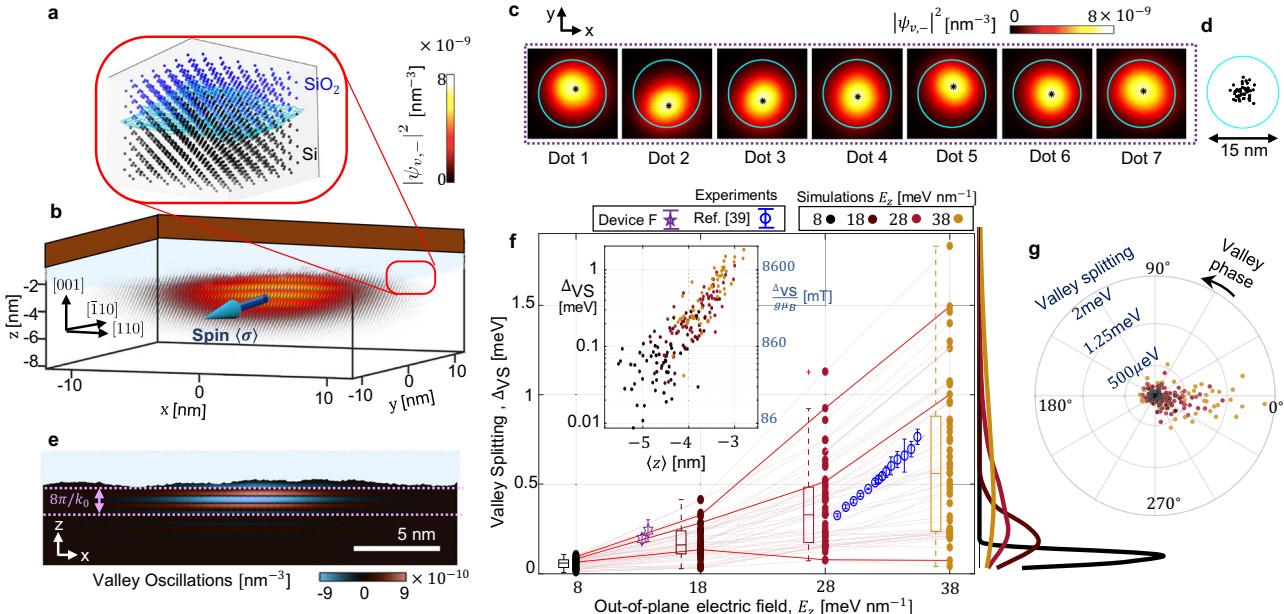

**Fig. 2 | Quantum dot variability. a** To simulate a Si/SiO$_2$ interface atomistically, we use a virtual lattice approximation (see Methods). This allows us to emulate the electronic properties of SiO$_2$—which does not have a regular lattice structure—in a simulated material with the same lattice structure as silicon. We then define the interface between the silicon lattice and its oxide with a simulated rough surface dividing the atomic sites between the two materials. **b** Three-dimensional visualisation of a CMOS quantum dot at the Si/SiO$_2$ interface simulated atomistically. The blue arrow represents the vector spin $\langle\boldsymbol{\sigma}\rangle$ averaged across the spin-orbitals of all the atoms in the quantum dot. **c** In-plane visualisation of the variability in the 7 quantum dots inside the purple rectangle in Fig. 1b. The 5-nm-diameter cyan circle is a static reference to compare the wavefunctions in different simulations. Black asterisks represent the centre of each quantum dot $\langle\mathbf{r}\rangle = \langle\psi|\mathbf{r}|\psi\rangle$. **d** Variability distribution of dot centres. **e** Visualisation of the valley oscillations parallel to the [001] lattice orientation. (see Methods section). The oscillation wavelength is $\frac{4\pi}{k_0}$, where

$k_0 = 0.82\frac{2\pi}{a_0}$ is the wavevector of the conduction band minima in the silicon crystal. **f** Valley splitting distribution of the 49 quantum dots versus the electric field. Box plots indicate the median (middle line), 25th, 75th percentile (box) and 5th and 95th percentile (whiskers) as well as outliers (single points). We compare our results with experimental data measured in device F and with measurements in ref. 39. Error bars indicate the standard deviation for the measured value. The electric fields are obtained from COMSOL simulations. The inset figure shows the correlation between the logarithm of the valley splitting versus the centre of the dot in the z-axis. The valley splitting is reconverted to magnetic field units in the right axis to compare it with the Zeeman splitting at different fields. **g** Distribution of valley phases versus the valley splitting. For convenience, we define $\phi_\nu = 0°$ as the point with the highest density of valley phases. The colour code represents the value of $E_z$ as in (**f**). Source data of (**f, g**) are provided in the Source Data file.

porate the self-affine characteristic of the Si/SiO$_2$ interface in our simulations.

In this paper, the quantum dots are simulated in a 7 × 7 grid array. Other architectures are also being explored[33], in part, because the practical design of such a dense array requires a sophisticated fabrication process with multiple metal layers to route the signals to the gates[2]. While dense wiring in multiple metal layers is routinely integrated into front-end-of-line industrial processes, qubit demonstrations using this integration have only recently been explored[34]. Moreover, this dense array leaves no space for interspersed readout devices such as single-electron transistors, and would be dependent on a gate-based readout approach[35] or would require quantum information to be shuffled to the edges of the array for readout[36]. Our results for qubit variability are not drastically affected by the choice of a grid array, except for a small degree of nearest neighbour correlations (see Supplementary Fig. 6), which once simulated across the full 450 × 450 nm Si/SiO$_2$ computer-generated interface, provides us with sufficient sampling to obtain statistical analyses accurately.

To understand how this roughness affects the quantum behaviour of electrons, it is necessary to focus on their wavefunction at the atomic scale (Fig. 2b). We use an atomistic tight-binding model of Si and SiO$_2$, which incorporates relativistic effects and the impact of a magnetic field, yielding eigenstates with realistic spin and valley structure. It can be used to calculate the ground state wavefunction and a few excited states.

Despite SiO$_2$ not having a regular lattice structure, we simulate it by assuming an atomistically ordered virtual crystal approximation (see Methods). The material is endowed with the same lattice structure as Si, and the tight-binding parameters are set to emulate the electronic structure of the interface[37,38]. This approximation allows us to simulate interface disorder atomistically, as seen in Fig. 2a. In addition, we developed techniques to extract this structure and calculate properties of the disordered quantum dot that would not be obtained with purely spectral analysis, such as the valley phase and the spin g-tensor (see Methods).

We find that the geometry explored here always leads to the successful formation of quantum dots, regardless of the local roughness profile. This is consistent with the yield of measurable quantum dots – all devices with functional gate electrodes (as determined by their influence on the charge-sensing single-electron transistor) could form controllable pairs of dots. Roughness mostly alters the quantum dot effective shape and centre position (Fig. 2c), with the location of the electron departing from the potential minimum by less than 5 nm, with a standard deviation of 1.4 nm (Fig. 2d). The dot position in the geometry studied here is highly tunable by biasing lateral gates (-5 nm/V, see Supplementary Fig. 2 and Supplementary Table 2), so that this disturbance can be corrected.

The excited orbital states are more impacted by the interface roughness. We are particularly interested in the first excited state, which corresponds to a valley excitation for a [001] Si/SiO$_2$ interface. The conduction band valleys along $\pm z$ crystal directions are

energetically favourable due to the effective mass anisotropy, and the degeneracy between these two valleys is lifted by the sharp interface. The performance of spin qubits is strongly impacted by the interface-induced valley coupling, which creates a superposition between the two valley quantum states. This superposition creates an oscillatory behaviour at the atomic scale, which can be seen in simulations in Fig. 2e. These oscillations are known to cause variability in valley structure even for interfaces with low levels of disorder. We refer to the relative phase between valleys in this superposition as valley phase and the energy separation between the two states as valley splitting.

To have pure spin systems, valley splittings exceeding the Zeeman energy are desirable. In Fig. 2f, we show how the valley splitting can be controlled by tuning the vertical electric field $E_z$, comparing measurements in two devices and the results of the simulations. The surface roughness causes variability in valley splitting of over one order of magnitude for a fixed electric field. The full range of valley splittings spreads from tens of μeV to a few meV, compatible with observed experimental values in CMOS devices[27,39].

When comparing these valley splittings to the spin splitting, we may ignore the variability in Zeeman energy (which is only of a few parts per thousand). Therefore, if we set relatively high electric confinement ($\approx 28$ meV nm$^{-1}$ is sufficient in our simulation) and we tune the magnetic field low enough (<700 mT in our study), all the 49 qubits in the simulation will obey the condition of valley splitting larger than the Zeeman splitting (See inset Fig. 2f). However, the fitted distributions in Fig. 2f show that the valley splitting can sometimes be very small, even at high electric confinements. This is an exceptional event, and for low magnetic fields, none of the 49 dots simulated here had valley splittings too small. However, it could potentially affect the development of large-scale quantum processors with millions of qubits as it is expected that a number of quantum dots will have the valley splitting clashing with the spin splitting. A possible solution would involve changing the number of electrons in the dot[40]. In worst-case scenarios, the dot must be discarded from the processor at the firmware level. A thorough discussion of the impact of this decision on error correction is presented in ref. 41.

Even with a consistently high valley splitting, qubit performance can still be impacted by variations in valley phases between neighbouring dots. The electron density will present Bloch oscillations in the $z$ direction, which are barely visible in Fig. 1f and were enhanced in Fig. 2e by taking the difference between the electron densities of both valley states (Methods). Notice that the $z$ oscillations have the same phase across the whole dot instead of conforming to the roughness of the oxide. This implies that the valley phase is well-defined even in the presence of surface disorder. This phase has an impact on operations that involve two dots, such as electron tunnelling and exchange coupling, because it determines whether these valley oscillations interfere constructively or destructively[42,43]. Figure 2g shows the valley phases across the 49 simulated dots, revealing that dots with larger valley splittings (typically above 300 μeV) tend to have similar valley phases, near zero in our definition. We speculate that this behaviour is a consequence of most dots with high valley splittings being formed in a preferred atomic layer, with similar z (see inset Fig. 2f), such that they have aligned valley phases.

## Qubit frequency variations

Spin–orbit effects lead to variability in qubit frequencies of the order of ~100 peV, which appears in the form of a variable $g$ factor for the spin subject to an external magnetic field. This variation represents less than 1% of the qubit frequency. The mean value of the ratio of Zeeman frequency and external magnetic field $f_{Zeeman}/B_0 = g\mu_B$ is 27.9 GHz T$^{-1}$, with the differences occurring only at the order of tens of MHz T$^{-1}$. This is a particularity of silicon electrons, whose spin–orbit coupling is among the smallest in all quantum dot technologies. This provides CMOS spin qubits with special protection against disorder

and electric fluctuations. However, these very small g-factor variations are still important for qubit operations aimed at higher than 99.9% fidelity, and they are directly linked to the roughness of the interface at the atomic scale.

Interface-induced spin–orbit coupling has two flavours in Si/SiO$_2$ interfaces—Rashba ($\alpha$) and Dresselhaus ($\beta$)[25]. These two can be experimentally differentiated by measuring the g-factor dependence on the in-plane magnetic field orientation $\varphi$[24]. The dependence is sinusoidal with the form $g(\phi) \approx g_0 + \alpha + \beta \sin(2\phi)$, where we take $g_0$ to be the theoretical bulk g-factor $g_0 = 1.9935$ calculated from atomistic simulations, including relativistic spin–orbit effects. The difference between the frequencies of any two qubits (Fig. 3a), as well as the electric field dependence $dg/dV$ (Fig. 3b) have the same behaviour. All 12 qubits measured in devices A to E show behaviours consistent with this description. Qubits in the same quantum dot but with different electron numbers are considered different in the total count, as they have substantially different g-factors (see data from device A in Fig. 3a, b). This is most likely due to different exposition to the atomic profile of the interface for electrons in different valley states. In most cases, Dresselhaus dominates both the total spin–orbit effect and its variability—with the exception of the configuration (1,1) in device A (Fig. 3a). The Rashba coefficient $\alpha$ is typically one order of magnitude smaller than $\beta$ (Fig. 3c, d).

We explore theoretically this variability by extracting the g-tensor of electrons in disordered quantum dots from the eigenfunctions calculated by tight binding. The results of simulations, shown as solid lines in Fig. 3a, b, are then used to extract the dependence of $\alpha$ and $\beta$ on the vertical electric field (Fig. 3c, d). The Dresselhaus effect emerges from breaking the lattice inversion symmetry near an interface, which explains why it is strongly dependent on the electric field that confines the electron against the oxide (Fig. 3c). The interface-induced Rashba effect is also dependent on the electric field, but at a smaller scale (Fig. 3d). Notice that a couple of simulations escape the overall trend. These are valley-spin degeneracies, occurring in these simulations because some valley splittings clash with the Zeeman splitting (Fig. 2f) at the input magnetic field of 1 T. While these degeneracies can be used for fast electrical driving[44–46], they significantly deviate from the target parameters for pure spin qubits. In practice, it is possible to either tune the valley splitting out of this regime or reduce the magnetic field to reduce the probability of accidental degeneracies, as discussed in the previous section.

We can also observe in Fig. 3c that the Dresselhaus parameter $\beta$ is bounded between two extreme values for each electric and magnetic field. This can be understood by analysing the perfectly flat [001] interface model, which introduces a distinction between the two sublattices in the diamond structure[26]. Figure 3e shows that in this case the Dresselhaus parameter $\beta$ is maximally positive for one sub-lattice termination and inverts for the other—the reason for this inversion can be understood by viewing the [001] terminations in Fig. 3f. In comparison, a rough interface will contain terminations in both sublattices, and the value of $\beta$ will then lie between these two limits (see also Supplementary Fig. 3).

Strategies for qubit control need to be designed according to these results in order to tolerate the natural statistical dispersion in qubit frequencies introduced by the oxide interface. Individual addressability is a particular challenge. The most common strategy explored so far for addressing a specific spin qubit relies on exploiting this g-factor variability, and driving a variable microwave field in resonance with its Larmor resonance frequency in order to induce spin rotations[4]. However, the fact that the spin–orbit effect has a maximum natural spread results in frequency crowding at large qubit numbers Fig. 3g, making it hard to address a given qubit without impacting other qubits with similar frequencies.

Instead, a more scalable pathway relies on a global microwave field acting on all qubits simultaneously[13,14]. Qubits will be driven in

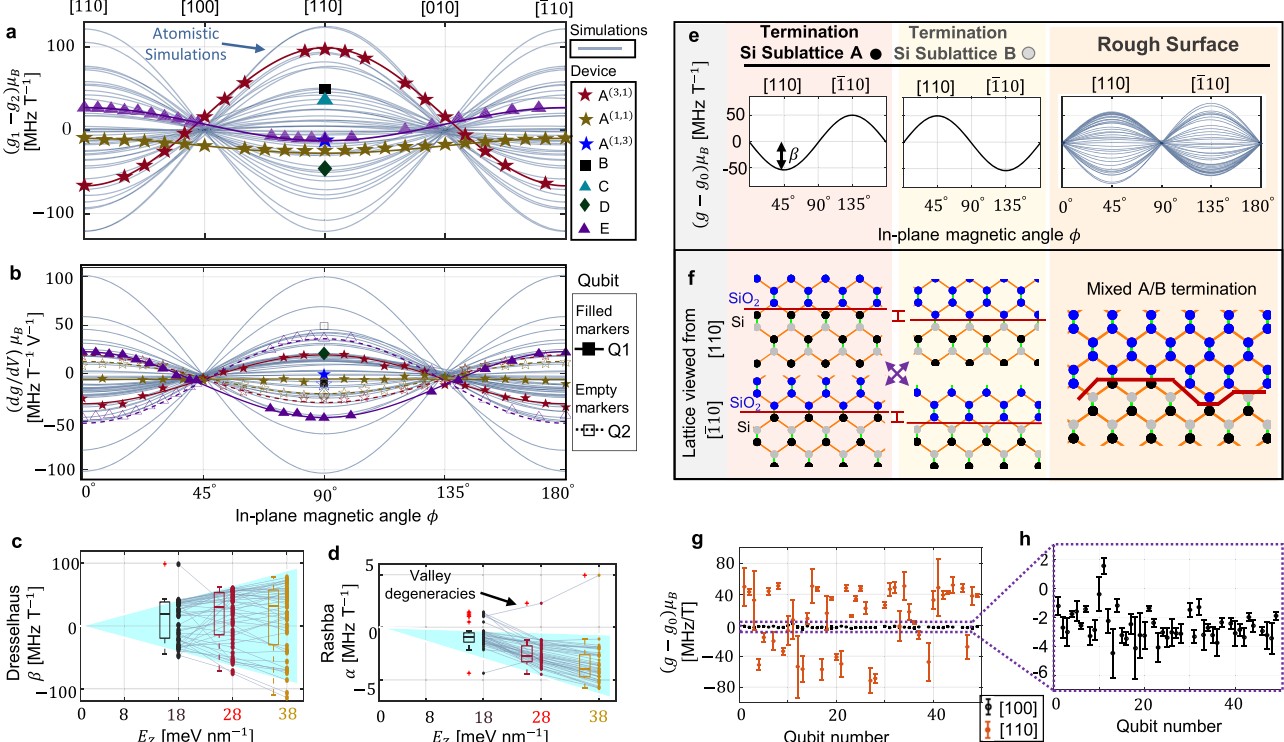

**Fig. 3 | Variability of the spin–orbit coupling. a, b** Comparison between g-factor variability in atomistic simulations and measurements in devices A to E under a varying magnetic field angle. On each device and configuration we measured two qubits. In **a** we compare the difference between the Larmor frequencies of the two-qubit qubits measured in each device $(g_1-g_2)\mu_B$ vs the differences between the frequency of neighbouring dots simulated atomistically (Fig. 1b). The marker for experimental data is associated with the device (A to E). In **b**, we compare the top gate Stark shift $dg/dV$ measured in the two qubits of each device, with atomistic simulations of the dots (methods). Two qubits in a device have a different Stark shift $dg/dV$ due to variations in surface roughness. Filled markers represent the data for the first qubit and empty makers for the second qubit (e.g. the empty purple triangles represent the Stark shifts measure on the second qubit of device E).
**c, d** Distribution of simulated Dresselhaus $\beta$ (**c**) and Rashba $\alpha$ (**d**) spin–orbit terms versus vertical electric field $E_z$. Box plots indicate the median (middle line), 25th, 75th percentile (box) and 5th and 95th percentile (whiskers), as well as outliers

(single points). **e, f** Schematic table showing that the sinusoidal dependence of the g-factors versus in-plane magnetic angle (**e**) follows from the anisotropy in the silicon lattice near the interface shown in (**f**). In an ideal flat surface, the border of silicon must end in one of the two possible sublattices A (black circles) or B (grey circles) and the interface looks different when observed from the [110] and the $1\bar{1}0$ lattice orientations. In a realistic rough surface, the border is a mixture of both A and B sub-lattice terminations, which explains the observed g-factor variability **g** Distribution of qubit frequencies for two magnetic field orientations: [110] and [100]. Qubits affected by near-valley degeneracies in the simulation are not included in this figure. The bars show an estimate for the maximum gate tunability of the g-factors with the top gate and a lateral gate (methods). The typical range of tunability for this gate is about 0.2 V for these devices. A higher potential bias could induce a charge transition, thus ruining the two-qubit system. **h** Zoom to the [100] data in (**g**). Source data are provided in the Source Data file.

resonance with the global field (forming a dressed qubit[47]), thus requiring that all qubits have identical frequencies aligned with the resonator frequency. Individual addressability is then achieved electrically, by tuning the qubits in and out of resonance via Stark shift. However, in the case of a magnetic field along [110], the range of electric control of the g-factors is insufficient to tune them into the same frequency, as can be seen in Fig. 3g. The variability in Larmor frequencies can be reduced by pointing the magnetic field along [100] (Fig. 3h) and by reducing the field magnitude to less than ≈100 mT[48]. These modifications lead to a standard deviation of less than 200 kHz, which is ideal for driving degenerate qubits with high fidelity (assuming a 2 MHz Rabi frequency in ref. 13). The Stark shift tunability also decreases, such that control strategies might need to cope with the inability to tune qubits completely out of resonance[13].

**The role of charge impurities in the presence of interface disorder**

Semiconductor devices are, in general, exposed to the presence of charge impurities in the oxide layer, originated from dangling bounds, Pb-centres, oxide vacancies, and other defects[16,21]. Some of these are fixed charged traps in the oxide, while others fluctuate over time, which is the origin of 1/f noise in semiconductor devices. The most

concerning charges from a variability perspective are charged traps. While two-level fluctuators are still important to understand qubit noise and decoherence, their absolute impact on the variability of qubit parameters occurs at a much smaller scale[48–50].

Previous works have modelled charged traps as negative electron charges $e^-$ directly at the Si/SiO$_2$ interface[28]. However, factors such as the positive correlation between the SiO$_2$ thickness with charge mobility in Hall bar devices[17,51], and measured large Dingle ratio[16] are possible signs that these charges might be dominantly located closer to the metal-oxide interface in some cases.

We simulated a uniform charge distribution distributed across bulk SiO$_2$ with density $-4 \times 10^{10}$ nm$^{-1}$[17] (see Fig. 4a). Each trap is assumed to lead to a deformation of the potential of

$$V_{\text{Trap}}(\mathbf{r}) = \frac{1}{4\pi\epsilon_{\text{Si/SiO}_2}} \frac{e^2}{|\mathbf{r} - \mathbf{r}_t|}, \tag{1}$$

where the trap at $\mathbf{r}_t$ and $\epsilon_{\text{Si/SiO}_2} = 0.5(\epsilon_{Si} + \epsilon_{SiO_2}) = 7.5\epsilon_0$. Here, we investigate the impact of these charges on spin qubits in the presence of interface disorder (Fig. 1b).

In most cases, in Fig. 4b, there are no charges trapped inside the dot region, so the quantum dot wavefunction is similar to the

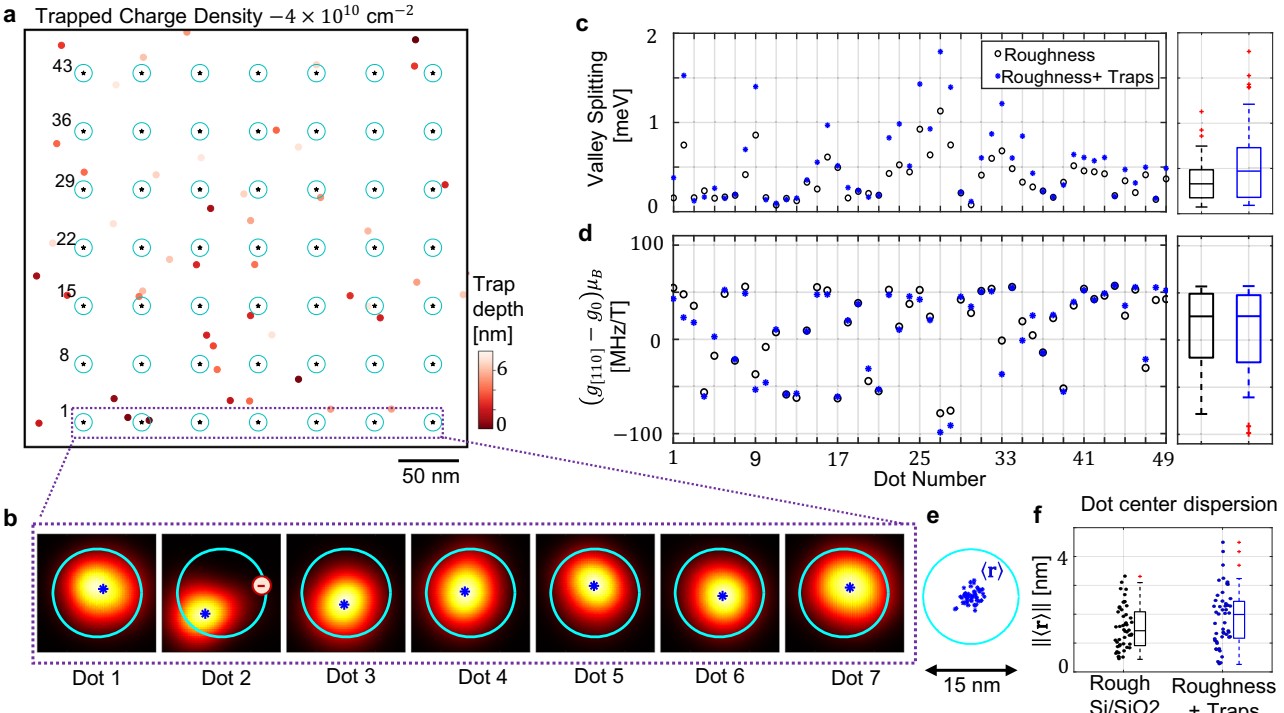

**Fig. 4 | Impact of charged impurities in quantum dots below a rough Si/SiO₂. a** A random distribution of negative charge impurities with typical industry-level densities of ~4 × 10¹⁰ nm⁻¹¹⁷. **b** Variability of the quantum dot wavefunctions of the same dots in Fig. 2c. There is a negative trap close to dot number 2. **c, d** Atomistic simulations of valley splittings (**b**) and g-factors (**c**) of each quantum dot in the 49 dot array, comparing simulations with and without charged traps. We set $E_z = 28$ meV nm⁻¹ in these simulations (Fig. 2f). In both cases, simulations are performed with the same surface profile (Fig. 1b). **e** Dispersion of dot centres under the presence of interface roughness and charged traps. **f** Comparison between the amplitude of the dispersion of dot centres in both scenarios. Box in all figures indicate median (middle line), 25th, 75th percentile (box) and 5th and 95th percentile (whiskers) as well as outliers (single points). Source data of **a**, **c**–**f** are provided in the Source Data file.

simulations without charged impurities (see Fig. 2c). The shifts in the potential profile induce small displacements of the quantum dots below the rough Si/SiO₂ interface. This leads to variations in valley spittings and g-factors with respect to initial simulations (Fig. 4c, d). An average increase in the valley splitting is observed, which is presumably due to an overall enhancement of the lateral confinement of the dots in the presence of the negative traps. This increase is typically larger for quantum dots that already had a large valley splitting (See Supplementary Fig. 7) due to their higher susceptibility to electric fluctuations. The dispersion of the dots centres is slightly larger (see Fig. 4e, f), but still small in comparison with the quantum dot size.

Only when the charge is inside the dot region, the quantum dot wavefunction is significantly affected (see, for instance, dot 2 in Fig. 4b). While the g-factor of this dot remains within range (Fig. 4c, d), the valley splitting enhancement is larger than what it is expected for typical values (See Supplementary Fig. 7). The probability of this effect happening is 5.6% for this particular charge density, as estimated from a Poisson distribution[52]. This situation could also affect large-scale architectures. However, it is unclear whether a qubit exposed to this type of defect would necessarily be unusable—the comparison between open and closed symbols in Fig. 4c, d indicates that the presence of traps does not increase the statistical dispersion of one-qubit parameters significantly. We will show next that the impact of traps on the two-qubit exchange coupling is also manageable with voltage offsets.

### Variability of exchange interactions

Besides the single qubit gate control, roughness and charge impurities also limit the homogeneity of the exchange coupling between neighbouring quantum dots. The differences in valley phase between dots, addressed in Fig. 2g, are the first source of variability in exchange coupling[42]. In the worst case, the valley phases would be random and the resulting exchange coupling would be impacted by the destructive interference of the valley oscillations. The probability of a completely destructive interference is, however, negligibly small. The typical valley interference causes, at worst, an offset in the exchange coupling of one or two orders of magnitude, which is easily corrected by an offset in the exchange control gate voltage, given that the tunability ranges from 6 to 10 decades per Volt. We find in simulations, however, that the disorder in the quantum dot position has a stronger impact.

The fact that exchange coupling is a contact interaction means that any effect impacting how the wavefunction tails off from one dot into its neighbouring dot, such as interdot distance and potential barrier height, has an exponential impact[53,54]. In the devices investigated here, the exchange is controlled by the action of the interstitial J-gate (see Fig. 5a). This gate induces a lateral displacement of the two dots toward each other, reducing the interdot distance at a rate of ~10 nm V⁻¹ (see Supplementary Table 2). This is contrary to the picture frequently evoked in this scenario, which assumes that the J-gate controls the electron penetration length into the classically forbidden region between dots without affecting much its position.

In Fig. 5b, we can see a method of extracting the exchange coupling by measuring the qubit resonant frequency for a randomly initialised pair of spins as a function of the J-gate voltage.

To simulate this system, we use a path integral Monte Carlo approach[55]. This method is relatively fast and intrinsically includes the effect of interactions in electron dynamics. For each exchange simulation, we sample realisations of likely paths of two electrons in a three-dimensional double quantum dot potential obtained from a finite elements simulation (see Fig. 5c). Interface roughness can be readily

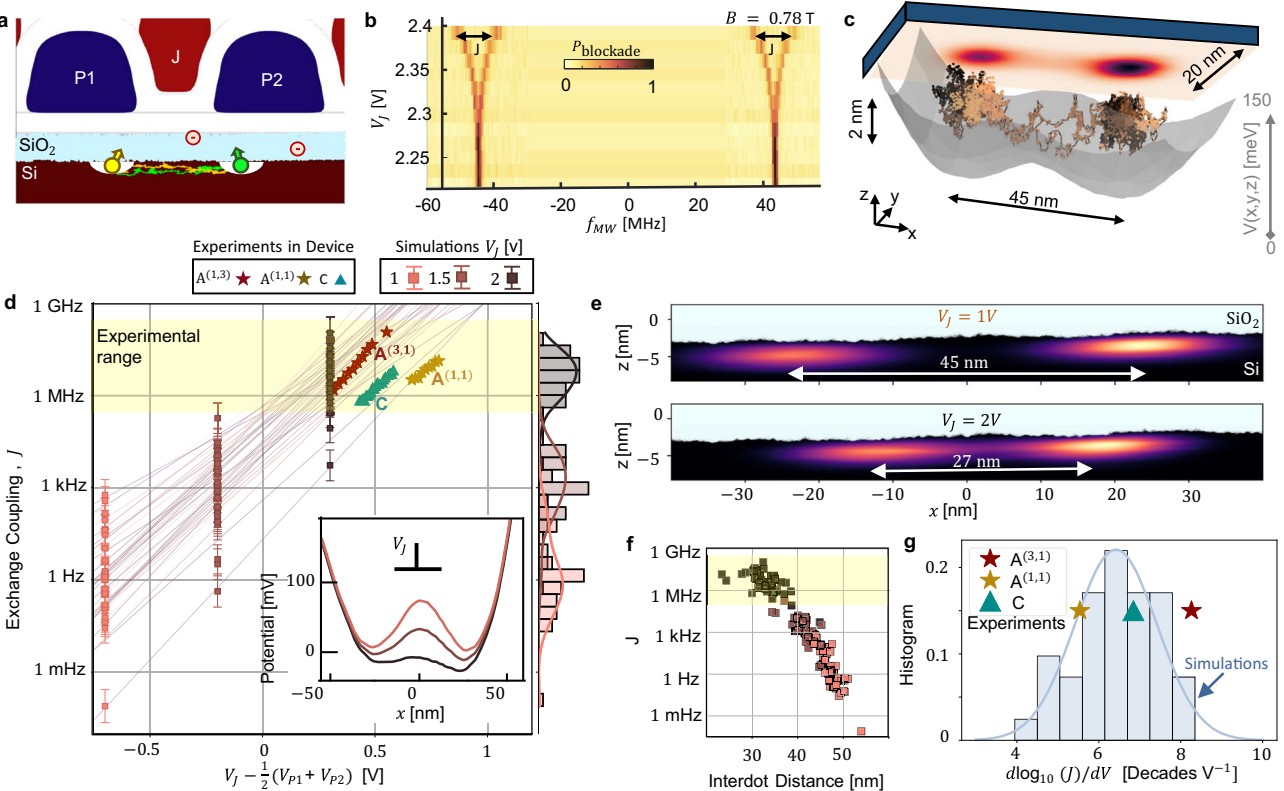

**Fig. 5 | Variability of the exchange coupling. a** Diagram depicting exchange interactions between neighbouring CMOS spin qubits in the presence of interface roughness and negatively charged traps. The green and yellow paths between the quantum dots represent the two electrons exchanging. These are simulated with a path integral Monte Carlo algorithm[57]. The interstitial exchange gate (J1) controls the interdot barrier. At high J-gate voltage, the spin interaction is high enough, and the two-qubit frequencies split. This is measured in device A at the (3 electron, 1 electron) configuration, (**b**). **c** Representation of the 3D path of the two electrons moving inside a double quantum dot generated with path integral Montecarlo (PIMC). The electric potential configuration is plotted in grey, and the electron density appears in the xy-plane. **d** Exchange versus J-gate detuning, comparing PIMC simulations in the presence of interface roughness (Fig. 1b) and charge traps

(Fig. 4a) with data from devices A and C. The colours of the simulations (small circular markers with error bars) represent the input potential values for the J-gate (1, 1.5, 2 V). Error bars are associated with the standard error of Path integral Monte Carlo simulations[57]. The inset figure shows a cut over the x-axis of the potential configuration for each value of J. This potential is obtained from finite element simulations in COMSOL MULTIPHYSICS (see Supplementary Fig. 2 and Methods). **e** Double dot wavefunctions simulated with path integral Monte Carlo (PIMC) for $V_J = 1\,V$ and $V_J = 2\,V$[57]. **f** Correlation between exchange coupling and interdot distance in PIMC simulations. **g** Histogram of the exchange controllability rates ($dJ/dV_J$) of device simulations, compared with experimental values (see markers). Source data of **b**, **d**, **f**, **g** are provided in the Source Data file.

included by defining a 3.1 eV step potential barrier to simulate the conduction band offset between silicon and the SiO₂ layer. The paths of these electrons are allowed to exchange between the dots, and the impact on the path action is used to estimate the exchange coupling[56]. The full method is described in ref. 57, and simulation details are included in the methods section.

In Fig. 5d, we compare the J-gate tuning of the exchange coupling with random surface and trap realisations against the experimental data obtained from devices A and C. We plot the exchange $J$ against $V_J - 0.5(V_{P1} + V_{P2})$, which is an approximate measure of the voltage bias between the J-gate and the plunger gates (our dots are fully isolated from the reference voltage set by the reservoir). All devices were tuned to the symmetric operation point[58].

Our dataset combines qubits at the outer shell of quantum dots with one electron and 3 electrons. The exchange interaction is larger for higher electron numbers, due to the increasing overlap between the wavefunction of the quantum dots. This is observed in device A, where the exchange baseline of the (3,1) configuration is higher than in (1,1). These situations are so far not considered in the simulations, which are performed in the (1,1) electron configuration. The gate stack also has a significant role in the exchange control, affecting the effectiveness of the electric fields generated by the J-gate in the

channel. Here, we compare devices A and C, which are both made with Pd/Ti gates with ALD oxides (see Table 1 and device architecture in Fig. 1f). Exchange control was also measured in devices E and F with Al gates in ref. 8, observing larger control rates and variability due to the absence of the ALD oxide and irregularities in the gate structure[18].

We found good agreement between exchange simulations and experimental data from devices A and C (Fig. 5d). An important part of this agreement was the inclusion of negatively charged impurities in the model, together with interface roughness (see Fig. 4a). The charges that most significantly impact the exchange coupling between neighbouring dots are the ones located in the interdot channel[57]. A single trap can reduce the exchange baseline a few decades. In total, this adds up to an average decrease of the exchange baseline of 1.5 decades in comparison with simulations without charge impurities, that have a higher baseline than experiments, as seen in Supplementary Fig. 5.

Importantly, the variability in exchange coupling can be compensated with more tunability. As seen in (Fig. 5e), the J-gate tunes the double dot potential, inducing a displacement of both quantum dots to the centre by almost 5 nm each. We can observe in Fig. 5f that this exchange dependence on displacement is consistent over multiple realisations, even when it is perturbed by the variability of the dot centres caused by the surface disorder and charged traps. Because of

the strong correlation between the interdot distance and the exchange coupling, we can associate the three orders of magnitude of exchange variability to these shifts in the interdot distance. Besides, the J-gate controls the interdot distance at an approximate rate of 10 nm per Volt, resulting in tunability ranges from 5 to 8 decades per volt in simulations and experiments (Fig. 5g). This is large enough to compensate for the disorder and consistently hit a target "on" exchange rate, as indicated by the yellow region in Fig. 5d.

## Discussion

The design of scalable quantum processor cells must be guided by a precise understanding of qubit variability. Besides demonstrating a complete strategy for diagnosing qubit variations for a given choice of qubit design, fabrication process and materials, this study leads to some general conclusions about the general physics of spins under Si/ SiO₂ interfaces. The main conclusion is that most of the qubit variability in current devices is explained by the roughness of the Si/SiO₂ interface roughness. The presence of charge impurities is also significant in regard to two-qubit properties. Other effects (such as strain inhomogeneity and geometrical deformation of the gates) can be mitigated down to levels that are, at most, comparable with these intrinsic mechanisms.

Another conclusion is that electric tuning of qubit frequencies (using spin–orbit effect) and exchange coupling (using barrier gates) both rely to a large extent on moving the quantum dot laterally, dragging it against the rough interface. This may lead to considerations in future designs of quantum dots and the methods for characterising the interface.

Charge noise coming from two-level fluctuators (TLFs) can also couple to the qubits through a similar mechanism. The induced displacements of the dots in the rough interface lead to small g-factor variations that can affect important qubit metrics, such as phase coherence $T_2^*$ [24,49]. The charge noise couples directly to the qubit Stark shift, so the methods of this paper can be applied to investigate the microscopic nature of this effect [59].

One remaining question is how much improvement can be realistically expected in the interface quality, and how it impacts the qubit performance. We address the last question in Supplementary Fig. 4, showing that the main benefits would be an enhancement of the average valley splitting and a smaller exchange variability. The spin–orbit coupling is not significantly affected due to its intrinsic atomistic dependence (Fig. 3f). Methods to improve the interface quality would involve replicating previously observed correlations between roughness amplitude and different fabrication parameters, such as growth time, oxide thickness, etc [60]. The characterisation methods developed in this study can help assess if the improvement in roughness amplitude occurs at the length scales that are more relevant for CMOS quantum processors.

This work focused on the case of Si/SiO₂ interfaces, which has been studied enough that we are able to draw firm conclusions on its impact on qubits. Other oxides or dielectrics might also be understood by adapting the methods developed here, providing pathways to shortcut the qualification of material stacks for quantum processor fabrication with the assistance of theoretical calculations.

Finally, this study realistically sets the ultimate variability of spin qubit parameters in CMOS devices. We may extract, for instance, the voltage offset that would be necessary to bring a qubit parameter to approximately the same value for all qubits in the architecture, which we call the voltage offset deviation VOD(x) for each parameter x (see Methods section). For example, the typical voltage offset to bring valley splittings to the same range is 0.58 V, while doing the same for g-factors requires 0.23 V if the magnetic field is pointed towards [100] and 9.1 V for a field along [110]. The smallest value is for the exchange coupling variability, which can be corrected with only 0.09 V voltage offsets. That clearly reveals that some parameters can be electrically

tuned to a target system-wide value while others will require the implementation of strategies to circumvent the variability with a combination of locally generated control signals and robust global control strategies [13]. These results outline the minimum demands for a CMOS spin qubit architecture.

## Methods

### Fabrication process
The SiO₂ gate oxide (7.5–8.0 nm) was thermally grown on the silicon surface in a custom-built high-quality oxide furnace as part of a standard MOS device fabrication process. The gate fabrication process was iterated multiple times to improve yield, which was an enabling feature for this study, leading to the successful formation of several devices with nominally identical layouts.

### Spin spectroscopy
To measure the Stark shift and the difference in Zeeman splittings between the spins, we have to be able to measure the Larmor frequency of the two qubits at a given operation point. In these experiments, we have double quantum dots, which we use as two electron spin qubits. To begin, we initialise both electrons in the same dot, forming a singlet state. We then separate the electrons, and depending on the rate of the separation, they will either end up in a T− state or having an odd parity, for instance, an up-down state. To find the Larmor frequency, we apply a fixed pulse or adiabatic microwave pulse with an antenna or resonator [61]. This pulse will flip a spin only if the applied frequency corresponds to the Larmor frequency. If the spin is flipped, the parity of the two spins will change too. We then bring the two electrons to a position where they are allowed to tunnel to the same dot only if their total parity is odd. If the parity is even, the electrons stay in their respective dots. We call this the Pauli-spin blockade-based parity readout [62,63]. The resulting charge state is read using a nearby charge sensor. Here, we used a single-electron transistor. The frequencies where we measured a flip in the parity of the two-spin states will correspond to the Zeeman splitting of one of the two qubits.

### Measurements of valley splitting
A tunnel rate-based spectroscopy method is used as an approximate measure of the valley splitting of quantum dots. The technique is as follows: (1) A repeated square-wave is applied to a dot gate, centred around a dot-to-reservoir charge transition. (2) The frequency of the square-wave is set equal to or faster than the dot-to-reservoir tunnel rate. In this mode, an electron will move in and out of the quantum dot in sync with the square-wave, but for only a proportion of the square-wave repetitions due to the tunnel rate. (3) The amplitude of the square-wave is swept from zero to 20 mV. The increase in amplitude causes excited state transitions to be accessed from the reservoir, which typically have increased tunnel rates to the quantum dot. (4) The changes in tunnel rate are fitted, and the splitting in amplitude between the ground state transition and excited state transition is multiplied by the dot gate lever-arm to retrieve the excited state energy. Here we assume the first excited state to be the valley excited state. This measurement technique is also explained in detail in ref. 64.

### Si/SiO₂ characterisation of interface from TEM images
We characterise the Si/SiO₂ roughness from TEM images of devices T1 to T6 (see Supplementary Fig. 1) using a similar procedure to [31]. To filter the interface, we apply image convolutions that enhance the differences between both materials. We then perform a power spectral density (PSD) decomposition of the filtered interface

$$\mathrm{PSD}^{1D}(\lambda) = \mathcal{C}_0 \left( \frac{2\pi}{\lambda} \right)^{-1-2H} \tag{2}$$

and confirm that the scaling of the Si/SiO$_2$ roughness is characteristic of a fractal self-affine interface[30,32].

We obtained an average Hurst exponent of $H = 0.28 \pm 0.2$ and a roughness amplitude parameter of $C_0 \approx 1.4$nm$^3$ for our devices. The PSD profiles can change for TEMs taken at different regions of the same device. This is normal as some line versions of a 2D surface can be smoother than others, and this behaviour is observed even in a surface generated numerically. We found that comparing average 1D parameters from multiple TEM images accounts for a better estimate of the global 2D profile (Supplementary Fig. 1).

## RMS characterisation of Si/SiO$_2$ interface from TEM images

To compute the scaling of the RMS($\lambda$) over a certain interface length characterised by the wavelength of its Fourier decomposition $\lambda$, we took separated segments of the fitted line surface from the TEM image of width $\lambda$ and computed the average RMS for each device. If a TEM has a width of $L = 40$ nm, we would compute the RMS($\lambda$) for $\lambda = L/2, L/4, \ldots$ until reaching the atomic scale $a_0$[30]. We can select more subsegments when $\lambda$ is small, which explains the smaller uncertainty of the average RMS for small $\lambda$. We found that the RMS scales with an exponent of $-H$ as

$$\text{RMS}(\lambda) = \frac{C_0}{4\pi H}\left(\frac{2\pi}{\lambda}\right)^{-H}, \tag{3}$$

which is expected for the PSD profile in equation (2)[32].

The RMS roughness amplitude of the computer-generated surface was obtained by averaging the results over line segments of the 2D surface profile, as in Supplementary Fig. 1c with the power spectral density.

## Random surface generation

The computer model of the surface in Fig. 1b was generated with a Fourier-filtering algorithm implemented in Matlab[65], that takes as input the 1D PSD profile in equation (2) and outputs a random rough surface with similar spectral density. The output surfaces look visually similar to the ones in the TEMs. To calibrate the model, we compare the average 1D PSD and 1D RMS scaling profiles, with the edge profile in the TEM images until we find a good match (Supplementary Fig. 1).

## Modelling of the digital twin of the devices

We create a 3D structure of the device in Matlab with the software provided by the DFX library. We begin by using a physical quantum dot electron-beam lithography (EBL) design layout as the primary framework and construct it as a 3D model that closely resembles its appearance in SEM and TEM images. Our software takes into account the levels of oxidation and thermal expansion that occur during the fabrication process to finely imitate the device geometry. We tweak these variables until the 3D model looks similar to transversal and in-parallel views of SEM and TEM images.

## Electrostatic simulations and quantum dot model

We import the digital twin device into COMSOL Multiphysics and perform electrostatic potential simulations with the integrated Poisson solver. We fit this to a harmonic model with a vertical electric field

$$V(x,y,z) = c_x(x - x_c)^2 + c_y(y - y_c)^2 + zE_z, \tag{4}$$

where $(x_c, y_c)$ is the centre of the parabolic potential, $E_z$ is the electric field in the z-axis (typically 8 to 40 meV nm$^{-1}$) and $c_x$, $c_y$ are the lateral curvatures (-0.3 meV nm$^{-2}$). We simulate potential sweeps from different gates to characterise their impact on the quantum dots (see Supplementary Fig. 2). The harmonic model allows to transform these actions to more intuitive parameters such as shifts in the electric

confinement, dot movement, ellipticity, etc. A summary of this impact is included in Supplementary Table 2.

## Atomistic simulations with interface roughness

We perform tight binding simulations in NEMO3D in the $sp^3d^5s^*$ 20-band model for Si, which intrinsically includes spin−orbit interactions[66]. In all simulations, the magnetic field magnitude is set to 1T, and we only vary the magnetic field orientation. To include surface disorder, we terminate the silicon lattice with the local section of the rough surface in Fig. 1b. We then label all the lattice sites above (below) the surface as SiO$_2$ (Si), as observed in Fig. 2a. The SiO$_2$ region is modelled with a $sp^3$ tight-binding (TB) model under a virtual crystal approximation (VCA). The SiO$_2$ TB parameters are optimised to reproduce the electrical properties of the oxide, namely the bandgap of 8.9 eV, conduction band offset of 3.15 eV relative to silicon, and conduction effective mass of 0.44m$_0$. The VCA model in TB assumes a well-defined crystal structure, in this case zincblende with a lattice constant having the same value of Si, but treats each atom as a fictitious SiO$_2$ atom. This is a standard way to model alloyed (SiGe) or disordered (SiO$_2$) materials under the VCA in the atomistic TB technique. At the interface region whether an atom is marked as a Si atom or SiO$_2$ atom then creates the atomistic disorder profile. The details of the model with parameter values can be found in refs. 37,67.

By loading the surface roughness profile into NEMO3D, we are able to simulate quantum dot wavefunctions with atomic resolution under the correct local symmetries induced by the disordered surface. A limitation of the model is that we cannot simulate atomically disordered Si-O bonds in the oxide. Considering the various geometrical permutations of such bonds in an amorphous solid, it becomes a computationally challenging problem for large-scale simulations. However, the VCA model does replicate the bulk electrical properties of SiO$_2$. Despite this limitation, we are able to simulate effectively a Si-SiO2 interface that, in general, is hard to describe from a tight binding approach.

## Valley Phase calculation with atomistic simulation

The atomistic tight binding software outputs the electron densities of the two valley states ($\| \Psi_{v_-}(x,y,z)\|^2$ and $\| \Psi_{v_+}(x,y,z)\|^2$)[68]. Their difference may be interpreted in terms of an envelope function $\Psi_{\text{Env}}$ multiplied by cosine oscillations with the wavevector of the conduction band minima $k_0 = 0.822\pi/a_0$

$$
\begin{aligned}
\psi_{\text{Osc}}(x,y,z) &= \left\|\Psi_{v_+}(x,y,z)\right\|^2 - \left\|\Psi_{v_-}(x,y,z)\right\|^2 \\
&\approx 2\Psi_{\text{Env}}(x,y,z)^2 \cos(-2i\kappa_0 z + i\phi_v).
\end{aligned}
\tag{5}
$$

A colour plot of $\psi_{\text{Osc}}$ is shown in Fig. 2e. To compute the valley phase, we average $\psi_{\text{Osc}}$ over the $x - y$ plane and perform a Fourier transform on the z-axis. There is a distinctive peak with frequency $2\kappa_0$, as it is expected for valley oscillations. The valley phase $\phi_v$ is obtained from the complex phase of the transform at this frequency.

## G-matrix computation from atomistic tight-binding

Atomistic simulations with NEMO3D include intrinsically spin−orbit interactions in the $sp^3d^5s^*$ 20-band model for silicon[66]. For all simulations, we set the magnetic field magnitude amplitude to 1 T and vary only the field orientation $\hat{B}$. The spin g-factor is independent of the magnitude of the magnetic field as long as the valley splitting is non-degenerate with the Zeeman splitting[44,46].

For any magnetic field **B**, the spin part of the Hamiltonian of the system is

$$H_{\text{Zeeman}} = \frac{\mu_B}{2}\boldsymbol{\sigma}^T \mathbb{G}\mathbf{B} = \frac{\mu_B}{2}\boldsymbol{\sigma} \cdot g_0\mathbf{B}_{eff}, \tag{6}$$

where $\mathbb{G}$ is the $\mathbb{G}$-matrix and $\mathbf{B}_{eff} = \frac{1}{g_0}\mathbb{G}\mathbf{B}$ is the effective magnetic field after including spin–orbit effects. The atomistic tight binding software outputs the Zeeman splitting $E_{Zeeman} = g_0\mu_B\|\mathbf{B}_{eff}\|$ and also the full wavefunction of the ground state $\Psi_\downarrow$ written in a base of atomic positions, orbitals and spins $|\mathbf{R}, \alpha, s\rangle$. From this, we can estimate the mean spin vector $\langle\boldsymbol{\sigma}_\downarrow\rangle = \langle\Psi_\downarrow|\boldsymbol{\sigma}|\Psi_\downarrow\rangle$. This spin aligns anti-parallel to the effective field $g_0\mathbf{B}_{eff}$ by definition. In total, we have obtained the magnitude and orientation of the vector $g_0B_{eff}$. If we perform this computation for three linearly independent magnetic fields $\mathbf{B}_1, \mathbf{B}_2, \mathbf{B}_3$, we will obtain the effective fields $\mathbf{B}_{eff,1}, \mathbf{B}_{eff,2}, \mathbf{B}_{eff,3}$. Then, the linear system $\mathbf{B}_{eff,i} = \mathbb{G}\mathbf{B}_i$ can be inverted to compute $\mathbb{G}$.

A typical G-matrix obtained from atomistic simulations is

$$\mathbb{G} = \begin{bmatrix} g_0 + \alpha' & \beta' & g_{13} \\ \beta' & g_0 + \alpha' & g_{23} \\ 0 & 0 & g_{33} \end{bmatrix}, \tag{7}$$

where the basis is aligned with the lattice orientations {[100], [010], [001]}. In here, $g_0 \approx 1.9937$ is the bulk g-factor, as obtained from the asymptotic behaviour of the simulations at low electric field. The entries $\alpha' \sim -10^{-3}$ and $\beta' \sim \pm 10^{-2}$ determine the in-plane spin–orbit-coupling and the parameters $g_{13} \sim \pm 10^{-3}$, $g_{23} \pm \sim 10^{-3}$ and $g_{33} \sim 2.00192 - \mathcal{O}(10^{-4})$ describe the out-of-plane components. The two remaining entries were calculated to be smaller than $10^{-5}$ at all cases and hence approximated to 0. To obtain the g-factor at any magnetic field orientation, we compute $g(\hat{r}) = \|\mathbb{G}\hat{r}\|$. If $\hat{r}(\phi) = [\cos\phi \sin\phi 0]^T$ is an in-plane normal vector

$$\mathbb{G}\hat{r}(\phi) = \begin{bmatrix} g_0 \cos\phi + \alpha' \cos\phi + \beta' \sin\phi \\ g_0 \sin\phi + \alpha' \sin\phi + \beta' \cos\phi \\ 0 \end{bmatrix}$$
$$:= \begin{bmatrix} g_x \\ g_y \\ 0 \end{bmatrix}. \tag{8}$$

Then

$$\|\mathbb{G}\hat{r}(\phi)\| = \sqrt{g_x^2 + g_y^2}$$
$$\approx \sqrt{g_0^2 + g_0\left(\alpha'^2 + 2\beta'^2 \sin(\phi)\cos(\phi)\right)} \tag{9}$$
$$\approx g_0 + \frac{\alpha'}{2g_0} + \frac{\beta'}{2g_0}\sin(2\phi)$$

under the approximation $\alpha', \beta' \ll g_0$ which is valid in this case. After replacing $\alpha := \frac{\alpha'}{2g_0} \approx \frac{\alpha'}{4}$ and $\beta := \frac{\beta'}{4}$, we recover the expression that led to the variability distributions in Fig. 3.

## Two electron Hamiltonian for path integral simulations

$$H_{2e}(\mathbf{r}_1(t), \mathbf{r}_2(t)) = \sum_{i=1}^{2} H_1(\mathbf{r}_i(t)) + \frac{e^2}{4\pi\epsilon_{Si}|\mathbf{r}_1 - \mathbf{r}_2|}, \tag{10}$$

where each single-electron Hamiltonian is given by

$$H_1(\mathbf{r}_i(t)) = \frac{\mathbf{v}_i^\dagger M_{Si}\mathbf{v}_i}{2} + V_{DQD}(\mathbf{r}_i) + V_{Si-SiO_2}\sigma(z_i - z_s(\mathbf{r}_i)). \tag{11}$$

Here $M_{Si}$ is the diagonal matrix $\text{Diag}(0.19, 0.19, 0.98)m_e$, denoting the effective mass of silicon electrons on each lattice orientation and $\mathbf{v}_i = \frac{d\mathbf{x}_i}{dt}$ is the velocity of each electron. $\epsilon_{Si}$ is set to $11.7\epsilon_0$. $V_{DQD}$ is the double quantum dot potential simulated in Comsol as in Supplementary Fig. 2.

The oxide interface is defined by a smooth step of $V_{Si-SiO_2} = 3.1\,eV$ with the function $\sigma(z) = \frac{1}{1 + e^{-4(z - z_s(\mathbf{r}))/a_0}}$. Where $a_0 = 0.543$ nm is the silicon lattice parameter. $z_s(\mathbf{r})$ defines $z$ coordinate of the rough surface at the position of $\mathbf{r}$ projected in the xy-plane $(r_x, r_y)$.

## Path integral simulation of exchange coupling

Hundreds of realisations of two-electron paths are sampled with path integral Monte Carlo (PIMC) with a Metropolis algorithm to minimise the partition function $\mathbb{Z} = e^{-S/\hbar}$[55], where $S$ is the total action $S = \sum_{m=0}^{N_t} \tau H_{2e}(\mathbf{r}_1(m\tau), \mathbf{r}_2(m\tau))$. We simulate both, paths that remain in separate quantum dots for the entire simulation and paths that exchange a few times between the quantum dots. The action of electron paths that exchange between the two dots is higher than for non-exchanging paths by an amount $\Delta S$, which allows to compute the exchange energy as $J = \frac{2}{\beta}e^{-\Delta S/\hbar}$[56]. As the estimates of the operators are computed from average sampled path realisations, the method provides natural error bars determined by the standard deviation of $-\Delta S$. To adapt the method to 3D electrons in MOS quantum dots, we had to do modifications to the main algorithm that are detailed in a separate paper (see ref. 57). For this simulations we used paths with $N_t = 8000$ time slices and $\beta\hbar = 4$ ps. The path integral method converges at these values, as shown in ref. 57.

## Voltage offset deviation (VOD)

$$\text{VOD}(\sigma) = \text{std}\left(\frac{\sigma - \langle\sigma\rangle}{d\sigma/dV}\right) \tag{12}$$

For any parameter $\sigma$ this metric estimates the deviation of voltage offsets that are needed to bring each variable to the average value $\langle\sigma\rangle$. For the main parameters discussed in this paper, we obtained:
1. Valley splitting: VOD(VS) = 0.58 V
2. g-factor [110]: VOD($g_{[110]}$) = 9.1 V Maximum among in-plane B-field angles
3. g-factor [110]: VOD($g_{[100]}$) = 0.23 V Minimum among in-plane B-field angles
4. Valley splitting: VOD(J) = 0.09 V

The values of $\sigma$ are obtained from roughness variability distributions. Only tunings in the range of hundreds of mV can be performed in real experiments. If the VOD is way higher than that, it means that it is not possible to tune $\sigma$ to the same value for all qubits.

## Estimating the tunability of qubit parameters

To compute the tunability of each qubit $d\sigma/dV$, we use simulations for different voltage tunings for the specific gates that have a significant impact on each one of the variables. For the exchange ($J$), we focus on the J-gate. In contrast, single qubit parameters like g-factors and valley splittings can be tuned with more than one gate. In here, we assume that the action is performed with a top gate and an additional lateral gate so that $d\sigma/dV = d\sigma/dV_{Top} + d\sigma/V_{Lat}$. At the same time, each of these tunings is divided as

$$\frac{d\sigma}{dV_{Top}} = \frac{d\sigma}{dE_z}\frac{dE_z}{dV_{Top}} + \frac{d\sigma}{dx}\frac{dx}{dV_{Top}}. \tag{13}$$

Estimates for the impact of different gates on single quantum dot parameters can be found in Supplementary Table 2. The values for $d\sigma/dE_z$ and $d\sigma/dx$ were simulated from small changes in the simulation parameters as in Figs. 2f, 3c, d.

## Data availability

The data for the figures generated in this study are provided in the Supplementary Information/Source Data file. Additional data and corresponding analysis code relevant to verify the results obtained

here and generate the plots shown in this paper have been deposited in the Figshare database under accession code https://doi.org/10.6084/m9.figshare.23507439. Source data are provided with this paper.

## Code availability

The path integral Monte Carlo algorithm to estimate exchange interactions is described in ref. 57. The algorithm to perform atomistic tight-binding simulations is described in ref. 66. The original code used for processing and generating all the plots in this paper have been deposited in the Figshare database under accession code https://doi.org/10.6084/m9.figshare.23507439.

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

## Acknowledgements

We acknowledge support from the Australian Research Council (FL190100167 and CE170100012), the US Army Research Office (W911NF-23-1-0092), the US Air Force Office of Scientific Research (FA2386-22-1-4070), and the NSW Node of the Australian National Fabrication Facility. The views and conclusions contained in this document are those of the authors and should not be interpreted as representing the official policies, either expressed or implied, of the Army Research Office, the US Air Force or the US government. The US Government is authorised to reproduce and distribute reprints for Government purposes notwith- standing any copyright notation herein. J.Y.H., S.S., M.K.F. and J.D.C. acknowledge support from the Sydney Quantum Academy. This project was undertaken with the assistance of resources and services from the National Computational Infrastructure (NCI), which is supported by the Australian Government and includes computations using the computational cluster Katana supported by Research Technology Services at UNSW Sydney.

## Author contributions

W.H.L. and F.E.H. fabricated the devices, with A.S.D.'s supervision, on isotopically enriched 28Si wafers supplied by K.M.I. (800 ppm), N.V.A., H.-J.P. and M.T. (50 ppm). T.T., W.G, J.Y.H, E.V, R.C.C.L. and S.S provided measurements of the qubit devices with the supervision of C.H.Y., A.S., A.L. and A.S.D. The data were gathered by J.D.C., who used it to estimate the CMOS variability in comparison with one-qubit and two-qubit simulations. J.D.C performed the one-qubit simulations with atomistic tight binding in NEMO3D with the supervision of R.R. and A.S. For two-qubit systems, J.D.C. performed the exchange simulations with an in-house Path Integral Monte Carlo code developed by P.Y.M, F.S and J.D.C. with the supervision of A.S. D.O coded an initial framework for the fractal analysis of the Si/SiO₂ interface that was later improved by J.D.C. with the supervision of A.S. F.E.H took the TEM images of the devices. J.Y.H. and D.D. coded the software that designs the digital twin of the devices with the supervision of C.C.E. and A.S. These digital twins were imported for electrostatic simulations in COMSOL by J.D.C. and D.D. with the supervision of C.C.E. M.K.F. provided theoretical support at all stages. J.D.C. and A.S. wrote the manuscript with input from all authors.

## Competing interests

A.S.D. is the CEO and a director of Diraq. WTT, W.G., E.V., C.C.E., A.L., C.H.Y., W.H.L., F.E.H., A.S. and A.S.D. declare equity interest in Diraq Pty Ltd. The remaining authors declare no competing interests.
