## [Peer Review File · Nature Communications]

REVIEWER COMMENTS

Reviewer #1 (Remarks to the Author):

Report on "Bounds to electron spin qubit variability for scalable CMOS architectures" by Jesús D. Cifuentes, et al.

This paper presents an in-depth analysis of the variability of electron spin qubits in Si/SiO₂, approached from both experimental and theoretical perspectives. Despite the significant amount of excellent research already published on electron spin qubits in Si/SiO₂, the effect of interface roughness on scaling up the number of qubits has always been a topic of debate. This paper provides some answers to this critical question.

The authors conducted an extensive series of experiments to assess the variability across different devices. The intersection of the theoretical work and experimental results is particularly appealing. It is notable that simulations can accurately predict valley splitting and exchange interaction, using the interface roughness obtained from TEM images as input parameters.

In conclusion, this study demonstrates that it is possible to anticipate the performance of electron spin qubits in Si/SiO₂ prior to conducting qubit experiments, by simply measuring the interface roughness. This might provide an insightful solution for scaling up the number of qubits in Si/SiO₂.

I have a couple of comments and questions that I would like to see addressed by the authors before I can recommend publication of this manuscript.

Specific comments:

-General question: Could the authors explain why the uniformity of the qubits is necessary? Can all the qubits not be tuned independently? How complex does the tuning process become with a larger number of qubits?

-If some of the devices (A-F) have appeared in previously published papers, could the authors explicitly state this?

-Fig.1a: It is unclear how to align all the electrodes (P gates and J gates) for 49 qubits. Are SETs still used for read-out at this stage?

-Fig. 1c: Could the authors define LCB and RCB?

-Fig. 1h: Why is the correlation length 500 nm for 49 dots, 100 nm for 4 dots, and 10nm for 1 dot?

-Fig. 1i: It is understandable how the authors obtained the PSD (Fig. 1h), but the process of obtaining the 1D RMS is unclear. Could the authors elaborate on this?

-Line 129: The citation of Fig. A2 is confusing here. How did the authors obtain the 1D RMS from Fig. A2?

-Lines 135-143: This part is unclear. Does a dot size of 10 nm correspond to one monolayer RMS? Does a separation between the dots of 50 nm correspond to 2 monolayers?

-Line 186: Does "the first one" refer to the first excited state?

-Fig. 2b: What does the blue arrow represent?

-Fig. 2g: Could the authors refer to it as an "inset"?

-Lines 214-221: What is the authors' message here? Is it easy or difficult to have a back gate?

-Lines 231-235: In the preceding lines, the authors stated that for 49 qubits all the valley splittings are higher than the Zeeman splitting. Could the authors explain why they expect that some of the valley splitting becomes lower than the Zeeman splitting as the number of qubits increases?

-Lines 236-237: If a qubit must be discarded, especially if that qubit is in the center of an array, doesn't this make performing quantum computation more challenging?

-Lines 257-258: Why do dots with larger valley splittings have similar valley phases?

-Fig.3a-d: The solid lines are difficult to discern.

-Fig. 3b: What are the open triangles? Where are black rectangles and open rectangles used?

-Fig. 3f: What do the green and black circles represent? Where are sublattice A and the sublattice B described? It might be beneficial to move Type A and Type B in Fig. A3 to Fig. 3f.

-Caption for Fig. 3e: Is a subject necessary here? "What" follows?

-Caption for Fig. 3f: What is meant by A (black) or B (gray)?

-Caption for Fig. 3g: How is the maximum gate tunability estimated?

-Line 260: Why should the variability in qubit frequency be small? Can different RF frequencies not be sent for different qubits?

-Line 294-295: From which figure can it be interpreted that the Rashba effect is dominant for one of the configurations in device A?

-Line 307: The strong dependence is not apparent in Fig. 3c.

-Lines 318-324: The bounds are not visible in Fig. 3c.

-Line 352: Could the authors explain why the variability is reduced for [001]?

-Lines 353-357: Why is a large Stark shift necessary? Are different Zeeman splittings required for different qubits?

-Fig.4a: What do the black arrows represent?

-Fig.4d: Which markers are from simulations? Which markers are from experiments? The solid lines are almost indistinguishable.

-Fig.4e: Could this be referred to as an inset?

-Fig.4h: Are the markers from experiments? Is the histogram from simulations?

-Lines 425-427: The authors found that the experimental exchange couplings have a lower baseline than their simulations. Is this in reference to Fig. 4d?

-Lines 464-470: I agree that the size of the quantum dots have major impacts but do the authors find that the size of the quantum dots have major impacts in this manuscript? If so, where?

-Fig. A4: How can the interface be made smoother experimentally?

Reviewer #2 (Remarks to the Author):

This manuscript reports on the effects of the Si/SiO₂ interface roughness on different spin qubit properties. From experimental TEM images and mathematical reconstruction, interface roughness is computed for different quantum devices. Then, the authors use tight binding simulations (including surface roughness) to compute the quantum dot structures, the excited states (including valley states), and the electron spin g-factor. In turn, a path integral Monte Carlo (derived in another paper) is utilized to study the spin exchange interactions in presence of surface roughness. Finally, the authors conclude that:

(i) the variability of current devices can be accounted for by the Si/SiO₂ interface roughness

(ii) a smaller dot size and gate pitches would decrease that variability

(iii) the electric tuning of the qubits must take into account the surface roughness (dot pulled and displaced along a rough interface).

The authors convincingly study the impact of the Si/SiO₂ interface roughness on the electron spin properties (valley splitting, excited spectrum, g-factor, SOI, spin exchange coupling controllability). Most of the presented simulations (core of this work) are in quantitative agreement with the experimental data.

However, the paper let aside the charge traps, which are, to my understanding, the major limitation of the Si/SiO₂ qubit physics and the semiconductor spin qubit in general (Martinez et al. Phys. Rev. Appl 17, 024022, 2022). More precisely, charge traps disturb the electrostatic landscape seen by a spin qubit. For a given (gate) electrostatic configuration, traps will affect the QD location and shape. Therefore, the very first step of prediction/tuning to make an array of quantum dots ready to hold spin qubits is strongly influenced by the presence of charge traps. Once the QD are formed and the spin qubit ready to operate, the typical signature of the presence of charge traps (beyond variability in lever arms) would be a change in the spin exchange coupling compared to a pristine device (which I believe is the origin of the discrepancy between the simulation and the experimental data of Fig.4d). Another signature will be the qubit charge noise.

When operating a quantum device, a complex sequence including slow/fast changes in either dot potential or spin exchange coupling are necessary. Here, I believe spin properties are computed in a static picture. I think that it would strengthen the paper to discuss these operations (which either modify the dot shapes, and more importantly here their locations) with the consideration of the interface roughness.

Are the simulations performed for a single electron per dot, or 3? The question stand for all simulations when compared to the experiment. Am I right saying that the 1,3 or 3,1 charge states were used to enhance the exchange coupling between the 2 qubits?

I find the concluding comment “smaller dots and pitches between dots restrict electrons to regions of the interface with smaller amplitude of the roughness, reducing the effects of valley interference” hard to understand. On one hand, closer spatial positions are more likely to have the “same z”, and therefore the phase difference in the valley state of 2 adjacent qubits are less likely to differ. On the other hand, bigger dots average out the roughness (to a certain extend), in the same fashion as for motional narrowing. A discussion on the valley phase difference as a function of the dot size and the gate pitch would help the reader understanding the authors’ conclusion.

I find the claim of lines 65-67 on the improvement of the fabrication processes letting the interface roughness the only property that is intrinsically limited very strong, especially from the perspective of charge traps (which could be discussed in terms of percolation density or charge noise).

I also have a list of minor comments, which could be addressed to clarify the discussion:

- Line 748, I believe [110] should be [100]
- Fig. 4g, I don't understand to what correspond the green line
- The definition of k_0 (given line 666) could also be written in the caption of Fig.2

To conclude, I think that this study, however convincing for the interface roughness, does not yet fit the wide scientific readership of Nature Communication. The discussion must include a part on the charge trap aspect of the materials considered, either based on experimental work or simulations.

If additional work on that aspect cannot be done, the scope of the paper must be narrower, or could include a discussion on the different operating regime of a quantum dot array (strong coupling for 2-

qubit gates, weak for idle and single qubit, etc.) and the impact of interface roughness on the qubit when “operating” an array.

Reviewer #3 (Remarks to the Author):

The manuscript "Bounds to electron spin qubit variability for scalable CMOS architectures" gives a detailed discussion on the influence on the Si/SiO₂ interface on the operation of spin qubits formed at this interface. Utilizing simulations and measurements the work presented assesses the viability for scaling of spin qubits formed with this architecture.

The work is timely and of high interest unfortunately the manuscript in its current state has, in my opinion, two overarching issues. First, the authors fail to contextualize their work within the wider field of CMOS compatible spin qubits. Second, I struggle to find explanations for a number of critical technical details in the manuscript and the references therein. I will go through the manuscript and highlight the specific paragraphs that I either cannot comprehend or disagree with.

In the introduction the authors state that "Enabling the breakthrough applications of quantum computation requires millions of qubits to perform error correction [7, 8], and it is infeasible to address all of these qubits with individualised pulses catering to their variable parameters." While this is a notion generally agreed upon in the field, the authors then got to simulate an array of 49 (7x7) qubits and compare to measurements on 12 qubits from 6 dual qubit devices. This raises the first technical questions: Why do the authors choose to simulate a 7x7 array rather than a set of 2 qubit devices in the same/a similar configuration as the fabricated devices? I cannot understand from the manuscript why the simulation of an array is beneficial in any way. Another issue that arises is that 49/12 qubits are nowhere near a few million. Note that a single failure in the simulations/measurements corresponds to 10.000s/100.000s of devices failing in a processor that contains a few million qubits.

The authors then go on to state "Variability may be caused by a few factors, such as strain, fabrication defects, accidental introduction of charged impurities in the oxide and so on [11–13]. Ultimately, most of these sources of variability can be improved with increasingly precise fabrication – industrial foundries focus most of their efforts in addressing these issues [6]. The only source of qubit variability that is inextricable to CMOS technology is the roughness of the interface between the crystalline Si and the amorphous SiO₂ ..." Reference 6 and 12 are papers discussing properties of CMOS spin qubits, reference 13 is a paper simulating the impact of strain and dopants on CMOS transistors, and reference 11 is a review discussing material requirements for Si based spin qubits. Unless the argument is that CMOS spin qubits have been manufactured and hence foundries are

doing a good enough job of controlling the process parameters, I do not see how these reference justify the authors statement that "most" variabilities can be improved by fabrication but the oxide interface is bound to stay as is. While it is true that CMOS manufacturing has made impressive strides, issues like the control of vacancies [<https://doi.org/10.1109/JPROC.2012.2189786>] and charges trapped on the oxide [<https://doi.org/10.1109/TED.2019.2907816>] have been issues for decades and may very well influence qubit variability. Furthermore, while the Si/SiO₂ interface is and always has been prevalent in CMOS devices, other oxides/materials have been evaluated [[10.1088/1674-4926/38/7/071002](https://doi.org/10.1088/1674-4926/38/7/071002), <https://www.mdpi.com/1996-1944/5/8/1413>] and to the best of my knowledge implemented for the production of memory devices [<https://www.mdpi.com/1996-1944/3/11/4950>, <https://doi.org/10.3390/nano13172456>]. Most importantly however, the authors completely ignore that CMOS compatible spin qubits have been successfully implemented using a buffer layer between the oxide and the charge carrier forming the spin qubit both in Germanium and Silicon in order to mitigate the issues arising from the oxide interface [<https://doi.org/10.1002/adfm.201807613>, <https://doi.org/10.1038/s41586-022-05117-x>, <https://doi.org/10.1063/5.0002013>]. Evaluating whether the introduction of the buffer layer is advisable could have been one of the major contributions of this work. Instead, other CMOS compatible qubit architectures are almost completely ignored by the authors. When addressing a wide, multidisciplinary audience as in Nature Communications this seems very careless.

It is then discussed how a 3D model of the Si/SiO₂ interface for tight binding simulations is built from the 2D TEM images. As already mentioned before it is not clear to me why the authors chose to simulate a 7x7 array. More importantly though, the authors then go on to model the amorphous silicon as a "virtual crystal approximation" using "SiO₂ atoms" on a silicon crystal lattice making the argument that this is standard in the field based on a single 10-year-old reference. This seems ill-advised. Real amorphous materials show variability in atomic configuration [[https://doi.org/10.1016/0022-3093\(76\)90023-5](https://doi.org/10.1016/0022-3093(76)90023-5)] that might very well influence both the local strain and charge density under the interface. Effectively ignoring the amorphous nature of the material when explicitly evaluating the interface between an amorphous and a crystalline solid seems fundamentally flawed. It certainly requires more justification than a single 10-year-old reference hidden in the Methods section. For example, a clear mention of this circumstance in the main manuscript and a paragraph discussing the potential shortcomings of this approach are in my opinion absolutely necessary.

Starting the comparison between the simulations and measured qubits, it is stated that "all devices with functional gate electrodes (as determined by their influence on the charge sensing single electron transistor) could form controllable pairs of dots." However, the yield is not mentioned. For the validity of the claim it is very relevant where 1% of all checked devices have "functional gate electrodes" or 99%. In this section two things are furthermore unclear to me:

- 1) I do not understand what is shown in Fig. 2e
- 2) In Fig. 2f, g the simulations are compared to 2 devices. Why only 2 and not all 6?

It is then stated that all 49 qubits in the simulation can be tuned to a valley splitting larger than the Zeeman splitting. However, this generic statement while crucially important, is not backed up with any further insights. Given that these are simulation results why do the authors not explicitly state/plot what the distribution of the valley splitting in the 49 quantum dots looks like? In principle it should be possible to evaluate the expected (best case) fidelity of the quantum bits from the calculations? Why is this not done? Given the information currently in the manuscript the statement that "a small but finite number of quantum dots that have valley splittings clashing with the spin splitting" which can be "discarded on the firmware level" seems unjustified. The authors provide neither the necessary statistics nor insights to substantiate such a claim.

In Figure 3 the authors evaluate the variability of the spin-orbit coupling of the different qubits. First, it is unclear what the magnetic field used for the simulations here (Fig.3 a) is (I assume still 700mT?). Second, I cannot find an explanation for g_1 and g_2 and hence don't understand what frequency difference g_1-g_2 refers to. Finally, it is stated that "All 12 qubits measured in devices A to E show behaviours consistent with this description." First, - A, B, C, D, E - are only 5 devices and hence only 10 qubits and not 12 or am I missing something here? Second, for 3 of the 5 devices only one measurement point seems to be shown, which is consistent with almost any description. It is then mentioned that a few simulations as well as one device (in a particular configuration) escape the overall trend and simply stated the this can be tuned away. However, in the introduction the authors explicitly state that "it is infeasible to address all of these qubits with individualised pulses catering to their variable parameters". So how is it feasible to tune the valley splitting of individual qubits? And if it is not, what are the implications for manufacturing millions of qubits? In addition, the authors find the "electric control of the g-factors is insufficient to tune them into the exact same frequency" and then conclude "Therefore, strategies for qubit control need to be designed to circumvent this variability and tolerate the natural dispersion in qubit frequencies introduced by the oxide interface." What strategies are the authors alluding to here? As best I can tell, one could also conclude that device architectures that avoid forming qubits at rough interfaces are more promising candidates for realizing spin qubits. As mentioned earlier, the authors consistently ignore alternative device architectures for no apparent reason.

In the final section the interaction between neighboring qubits is investigated. A path integral method is used. Again, a number of technical question arise that I cannot find an explanation for in the manuscript. Based on Figure 4c it looks like to electrons are moving in a parabolic potential. How is this potential generated? What justifies its symmetries? How is the amorphous nature of the oxide implemented? It is stated that the "Interface roughness can be readily included by defining a 3.1 eV step potential barrier to simulate the conduction band offset between silicon and the SiO₂ layer." I do not understand how the authors "readily" move from a physical roughness on the atomic scale to a potential barrier which almost certainly won't be atomically sharp due to the quantum nature of the electron wave function in the material stack.

The simulations are compared to 2 devices again. Why not all 12? How are the devices the authors choose for comparison selected? The results are that "the experimental exchange couplings have a lower baseline than our simulations despite a good agreement with the exchange controllability

rates". This disagreement is attributed to "valley interference in the devices". Again, then why are these 2 devices chosen for comparison if they are suspected to have valley interference? It was stated earlier that devices can readily be tuned out of valley interference? Did the authors attempt this? Would it also be feasible to conclude that the approximations made in the simulations might cause the disagreement between measurement and simulation?

The section concludes with the statement "In the specific geometry simulated and measured here, the tunability ranges from 6 to 10 decades per volt, large enough to compensate for the interface disorder and consistently hit a target "on" exchange rate across all devices." Does this mean all simulated or all measured devices? Given that this is a fairly important statement - where is this demonstrated or why is it not demonstrated?

In the conclusions the authors argue that "Finally, this study realistically sets the ultimate variability of qubit parameters." I do not understand this statement. The study investigated variability of qubit parameters of one isolated issue (Si/SiO₂ interface roughness) in a specific qubit architecture. It seems absurd to claim that this sets any ultimate limits on qubit parameters in general. In addition, the authors state "These results outline the minimum demands for an architecture that can deal with qubit variations while maintaining high fidelity." Similar to the previous statement this seems completely unjustified. Nothing in this work addresses qubits in general and as mentioned several times before, even when looking at CMOS compatible electron spin qubits in silicon, architectures that avoid the main/only source of variability investigated here are readily available and have been demonstrated experimentally.

In my opinion the manuscript is not publishable in its current state. The following issues would have to be addressed:

First, a number of overly general statements need to be adjusted.

Second, the choice of the specific qubits (chosen from the 12 available) used for comparisons to the respective simulations needs to be justified.

Third, the missing technical details outlined above should be addressed. In particular, approximations made in the respective simulations should be mentioned and their impact on the results should be discussed. Furthermore, all (free) parameters used in all simulations need to either be stated or sourced so that the work is reproducible.

Fourth, other CMOS compatible spin qubit architectures need to be mentioned and the authors need to justify the choice of the architecture investigated here.

After addressing these points, the manuscript would in my opinion be publishable in a specialized journal. In order to justify publication in a multi-disciplinary journal with a large audience like Nature Communications, the authors would have to furthermore show that their work has an impact on semiconductor spin qubit devices beyond the specific device architecture investigated in this work – i.e. answer the question why the impact of the Si/SiO₂ interface on qubit operation cannot simply be

bypassed by either the introduction of buffer layers or the use of other CMOS compatible (potentially crystalline or poly-crystalline) oxides.

Reviewer #1:

Report on "Bounds to electron spin qubit variability for scalable CMOS architectures" by Jesús D. Cifuentes, et al.

This paper presents an in-depth analysis of the variability of electron spin qubits in Si/SiO₂, approached from both experimental and theoretical perspectives. Despite the significant amount of excellent research already published on electron spin qubits in Si/SiO₂, the effect of interface roughness on scaling up the number of qubits has always been a topic of debate. This paper provides some answers to this critical question.

The authors conducted an extensive series of experiments to assess the variability across different devices. The intersection of the theoretical work and experimental results is particularly appealing. It is notable that simulations can accurately predict valley splitting and exchange interaction, using the interface roughness obtained from TEM images as input parameters.

In conclusion, this study demonstrates that it is possible to anticipate the performance of electron spin qubits in Si/SiO₂ prior to conducting qubit experiments, by simply measuring the interface roughness. This might provide an insightful solution for scaling up the number of qubits in Si/SiO₂. I have a couple of comments and questions that I would like to see addressed by the authors before I can recommend publication of this manuscript.

We thank Reviewer 1 for valuable feedback and for referring to our paper as appealing due to accurate predictions of experimental qubit data. We also consider this as the most important insight of the paper, as there is currently a gap in the field between theoretical predictions of qubit variability and observed experimental variations.

The reviewer made several interesting comments regarding the importance of qubit uniformity, small variability in qubit frequencies, and other aspects that are crucial for the technology. We answer these questions in this document and add more information in reference to these points on the main paper. We also thank the reviewer for finding some typos in figures and text, that have now been corrected.

1) General question: Could the authors explain why the uniformity of the qubits is necessary? Can all the qubits not be tuned independently? How complex does the tuning process become with a larger number of qubits?

This question speaks to the core of the architectural principles behind CMOS. Indeed, with the co-integration of electronics it is possible to independently tune qubits in a scalable manner, which would help cope with this variability.

In this sense, the key result in our work is exactly how much tuning would be required. Designing scalable control cells that can offset qubit properties across a large range and with sufficient accuracy is increasingly challenging if the electric tunability is too small or the range of variations is too large. The quantitative considerations in our study are key to the practical design of such systems, which need to otherwise optimise their footprint in both size and thermal budget.

We have added this discussion in the manuscript to provide more information about this:

Lines 45-58: “Instead, this variability must be embraced and corrected through the combination of on-chip electronics operating at cryogenic temperatures\cite{Veldhorst2017, vandersypen_interfacing_2017}, and robust quantum control pulses that will be shared among several qubits ~\cite{hansen_pulse_2021,hansen2023entangling}. **Designing control cells that can offset qubit properties across a large range and with sufficient accuracy is only possible if the electric tunability compensates for the range of variations. As we will discuss in this paper, this interplay between variability and tunability differs between qubit parameters (one-qubit frequencies and two-qubit couplings, etc), so a different control strategy must be adopted in each scenario.**”

Finally, we have highlighted this point in the main conclusions in the paper.

Lines 651-653: “The design of scalable quantum processor cells must be guided by a precise understanding of qubit variability”.

2) If some of the devices (A-F) have appeared in previously published papers, could the authors explicitly state this?

We added a new column in Table 1 including the papers that have been published based on data from each device.

Please note that the voltage configurations where the data was measured in the referenced publications might differ from the data included in this paper. In particular, the publication [Gilbert, W. et al. Nat. Nanotechnology. 1–6 (2023)] is focused on a particular regime where orbital degeneracies were intentionally created to enhance the spin-orbit coupling, which significantly affects the g-factor variability. Experiments and simulations are done outside of this regime as it is stated in the section of g-factor variability.

3) Fig.1a: It is unclear how to align all the electrodes (P gates and J gates) for 49 qubits. Are SETs still used for read-out at this stage?

We have not explicitly committed to any one form of gate layout for this work to avoid confusion between metal gate stack architecture and the role of the interface. An example showing methods to implement such square grid of quantum dots is discussed in Veldhorst, M. et al Nature Communications 8, 1–8 (2017).

4) Fig. 1c: Could the authors define LCB and RCB?

LCB (Left confinement barriers) and RCB (Right confinement barrier) are lateral barrier gates to confine the quantum dot in the y direction. We have now included the text explicitly in Figure 1c.

5) Fig. 1h: Why is the correlation length 500 nm for 49 dots, 100 nm for 4 dots, and 10nm for 1 dot?

We have included this text in the caption of Fig 1h for clarity:

“The data is plotted as a function of $\lambda = \frac{2\pi}{q}$, where q is the wavenumber. This allows us to compare λ with most relevant length scales, namely a_{Si} is the silicon lattice parameter $a_{Si} = 0.357$ nm, the dot radius of 10-15 nm, the double dot size of 80-100 nm and the full 500 nm of the simulation cell containing all 7x7 dots.”

Notice that previously we were referring to 4 dots (2x2) and 49 dots (7x7), because we were considering the grid architecture. We changed this to 2 dots and 7 dots in Fig. 1h as it makes the message clearer.

6) Fig. 1i: It is understandable how the authors obtained the PSD (Fig. 1h), but the process of obtaining the 1D RMS is unclear. Could the authors elaborate on this?

Answer in next question

7) Line 129: The citation of Fig. A2 is confusing here. How did the authors obtain the 1D RMS from Fig. A2?

We agree that Fig. A2 (now Supplementary Fig. 2) was wrongfully cited. We have eliminated this citation.

A paragraph about RMS characterization was included in the methods section of the first manuscript together with the PSD characterization. It was not very visible though, so we separated this part into a new section inside the methods:

Lines 829-842 :

“RMS characterization of Si/SiO₂ interface from TEM images

To compute the scaling of the RMS over a wavelength amplitude λ we took separated segments of width λ from the fitted line surface of the TEM image. We computed the average RMS for these segments. If a TEM has a width of $L = 40$ nm we would compute the $\text{RMS}(\lambda)$ for $\lambda = \{L/2, L/4, \dots\}$ until reaching the atomic scale a_0 . We can select more subsegments when λ is small, which explains the smaller the uncertainty of the average RMS for small λ 's. “

We added this reference in the caption of Fig. 1i to methods section.

“Average RMS of segments of length λ for the random surface generated numerically and for device T1 to T6 (**details in methods section**).”

8) Lines 135-143: This part is unclear. Does a dot size of 10 nm correspond to one monolayer RMS? Does a separation between the dots of 50 nm correspond to 2 monolayers?

We have rephrased this part to improve clarity.

Lines 202-206: “The root-mean-square (RMS) is approximately 0.15 nm within a dot (about 1 monolayer of the Si lattice), and almost twice for the double dot (RMS is 0.3nm, about 2 monolayers of the Si lattice).”

9) Line 186: Does "the first one" refer to the first excited state?

Yes, we are talking about the first excited state. We have rephrased this part for clarity.

Lines 257-260: “We are particularly interested in the **first excited state**, which corresponds to a valley excitation for a [001] Si/SiO₂ interface.”

10) What does the blue arrow represent?

It represents the spin vector averaged across all the atomic sites in the whole quantum dot. We include this text explicitly in Fig. 2b and add this in the caption of Fig. 2b for clarity.

“The blue arrow represents the vector spin $\langle \vec{\sigma} \rangle$ averaged across the spin-orbitals of all the atoms in the quantum dot.”

11) Could the authors refer to it as an "inset"?

We refer to it as an inset in the new text.

12) Lines 214-221: What is the authors' message here? Is it easy or difficult to have a back gate?

Lines 214-221: "In general, the field dependence of valley splitting is linear for small ranges of electric field variation. In the absence of back gates (such as those available in SOI devices [26]), the vertical electric field cannot be tuned independently of the quantum dot chemical potential, which limits the range of tunability of the electric field before a charge transition occurs. When comparing these valley splittings to the spin splitting, we may ignore the variability in Zeeman energy (which is only of a few parts per thousand)"

We thank the referee for pointing out that the lines copied above had an unclear role in the manuscript. We decided to remove this discussion entirely, since a more thorough description of this type of controllability would be unduly long and technical for such a journal as Nature Communications.

For the reviewer's own reference, back gates are easily incorporated in CMOS architectures based on fully depleted silicon-on-insulator technology, which is sometimes available in tier 1 foundries. This allows the experimentalist to enhance the vertical component of the electric field independently of the bias applied to the plunger gate regarding the ground. Significant developments in this direction were demonstrated by researcher at CEA-LETi in France, for instance. We refrain, however, from discussing this in the revised manuscript since non-experts would require a lot more explanation in order to fully appreciate this conclusion, and this discussion is not really impacting the overall scientific conclusions in our work.

13) Lines 231-235: In the preceding lines, the authors stated that for 49 qubits all the valley splitting are higher than the Zeeman splitting. Could the authors explain why they expect that some of the valley splitting becomes lower than the Zeeman splitting as the number of qubits increases?

We thank the reviewer for pointing out this. We agree that it required more information. We have added a lateral histogram of the valley splitting (Fig. 1e) and the following text:

Lines 296-301: "**However, the fitted distributions in Fig. 2f show that the valley splitting has chances to be arbitrarily small, even at high electric confinements. This is an exceptional event with probability smaller than 1/49.** However, it could potentially affect the development of large-scale quantum processors with millions of qubits as it is expected that a number of quantum dots will have the valley splitting clashing with the spin splitting.

14) Lines 236-237: If a qubit must be discarded, especially if that qubit is in the center of an array, doesn't this make performing quantum computation more challenging?

That is precisely right. This is exactly why we are concerned about qubit variability and the admissible range of variation before a qubit is considered too defective. A very thorough discussion of how the qubits in an array impact quantities like threshold and overheads is presented in Shota Nagayama *et al* 2017 *New J. Phys.* **19** 023050; (link: <https://iopscience.iop.org/article/10.1088/1367-2630/aa5918/meta>)

We have added a reference to this work in the new manuscript with a comment.

Lines 309-311: "A thorough discussion of the impact of this decision on error correction is presented in Ref.~\cite{Nagayama_2017}."...

15) Lines 257-258: Why do dots with larger valley splittings have similar valley phases?

Lines 257-258: "Revealing that dots with larger valley splittings (typically above 300 μeV) tend to have similar valley phases, near zero in our definition"

This result was unexpected for us. We were expecting the different interfaces at the different quantum dot sites to pin the valley phase to different values, essentially randomly.

Instead, we found that the valley phases only differed significantly in sites where the roughness led to a reduction in valley splitting. We have a few hypotheses why this might be true, but the scale of simulations required to test these hypotheses was prohibitively large for us to implement at this stage.

The critical aspect here is that all the 7x7 qubits were simulated over a single computer-generated surface, which then incorporated the long-range correlations of the roughness explicitly. Our main suspicion is that the form of the roughness that results from the fractal structure of the interface leads to a preferred atomic layer for the dots to form, which consist of the most common atomic layer where the silicon lattice gets terminated. This most common layer generates regions within the dot which maximize the valley splitting, therefore minimizing the total energy of the electron (even if the electron does not end up forming squarely in the centre of the dot). The cases where such a plateau does not exist would force the electron to form in any other point of the interface, exposing it to more roughness (which reduces the valley splitting) and to other atomic layer terminations (which would lead to changes in valley phase).

Unfortunately drawing a firm conclusion on this hypothesis would require one or two orders of magnitude more simulations in order to confirm this effect across multiple realisations of the interface roughness and form correlations between the presence of plateaus, the valley splitting and the pinning of the phase. While this is not computationally impossible, this adds a lot to the current study and therefore we considered it to be out of scope at this stage for our manuscript. The current study already contains approximately 1 million cpu*hours' worth of simulations.

We added the following paragraph to this part:

Lines 328 -336: "Fig. 2f shows the valley phases across the 49 simulated dots, revealing that dots with larger valley splittings (typically above 300 μeV) tend to have similar valley phases, near zero in our definition. We speculate that this behaviour is a consequence of most dots with high valley splittings being formed in a preferred atomic layer, with similar $\langle z \rangle$ (see inset Fig. 2f), so they have aligned valley phases."

16) Fig.3a-d: The solid lines are difficult to discern.

We thank the reviewer for the suggestion. We have improved the quality of the solid lines in Figure Fig.3a-d to improve visibility.

17) Fig. 3b: What are the open triangles? Where are black rectangles and open rectangles used?

We have modified the legend of Fig. 3b for clarity. All filled markers are data from the qubit 1 of each device. Empty markers are data from qubit two. The device is identified by the marker in Fig. 3a.

18) Fig. 3f: What do the green and black circles represent? Where are sublattice A and the sublattice B described? It might be beneficial to move Type A and Type B in Fig. A3 to Fig. 3f.

The grey (not green) and black circles are used to represent the two sublattices in the silicon FCC lattice. We have made Fig. 3f larger for clarity. We included grey and black circles in the top of the table to help in distinguishing which is the last sublattice type below the interface.

19) Caption for Fig. 3e: Is a subject necessary here? "What" follows?

We corrected this mistake in the caption for Fig. 3e:

"Schematic table showing that the sinusoidal dependence of the g-factors versus in-plane magnetic angle (θ) follows from the anisotropy in the silicon lattice near the interface shown in (f)."

20) Caption for Fig. 3f: What is meant by A (black) or B (gray)?

Sublattices A and B are the two sublattices in a silicon crystal. Sublattice A is represented by black circles and sublattice B by gray circles. We have added this information in the title of the table in Fig. 3f and included these modifications in the caption of Fig. 3f:

"In an ideal flat surface, the border of silicon must end in one of the two possible sublattices A (black circles) or B (gray circles) and the interface looks different ..."

21) Caption for Fig. 3g: How is the maximum gate tunability estimated?

This is a very important point. We have included the following text in the caption of Fig. 3g for clarity about this point:

"The typical range of tunability for this gate is about 0.2~V for these devices. A higher potential bias could induce a charge transition, thus ruining the two-qubit system.

22) Line 260: Why should the variability in qubit frequency be small? Can different RF frequencies not be sent for different qubits?

We thank the reviewer for this excellent question. Indeed, our description of this part was limited, mostly to save in text length. We have rephrased this section in order to add more information about this point.

Lines 423-446 "Strategies for qubit control need to be designed according to these results, in order to tolerate the natural dispersion in qubit frequencies introduced by the oxide interface. Individual addressability is a particular challenge. The most common strategy explored so far for addressing a specific spin qubit relies on exploiting this g-factor variability, **and driving a variable microwave field aligned with its unique Larmor resonance frequency in order to induce gate rotations** \cite{Veldhorst2014}. However, the fact that the spin-orbit effect has a maximum natural spread, **results in frequency crowding at large qubit numbers** (Fig. 3g) making it hard to address a given qubit without impacting other qubits with similar frequencies.

Instead, a more scalable pathway relies on a global microwave field acting on all qubits simultaneously~\cite{hansen2023entangling, hansen_pulse_2021}. Qubits will be driven in-resonance with the global field (forming a dressed qubit \cite{seedhouse_quantum_2021}), **thus requiring that all qubits have identical frequencies aligned with the resonator frequency.** [...]"

23) Line 294-295: From which figure can it be interpreted that the Rashba effect is dominant for one of the configurations in device A?

We included a reference to the specific figure and data point to clarify this part.

Line 379: "In most cases Dresselhaus dominates both the total spin-orbit effect and its variability -- with the exception of the **configuration (1,1)** in device A (**Fig. 3a**)"

24) Line 307: The strong dependence is not apparent in Fig. 3c.

Answer in next question

Lines 318-324: The bounds are not visible in Fig. 3c.

Lines: "The Dresselhaus effect emerges from breaking the lattice inversion symmetry near an interface, which explains why it is strongly dependent on the electric field that confines the electron against the oxide."

We have changed the colours in Fig. 3c. expecting that the boundaries of Dresselhaus parameter β become clearer (green shaded region). We think this might be the reason why the strong dependence is not apparent. Dresselhaus variability tends to 0 in the bulk limit ($E_z = 0$ meV/nm) and increases significantly with the electric confinement against the interface.

25) Line 352: Could the authors explain why the variability is reduced for [001]?

Line 352 "The reduced variability enabled by pointing the field along **[001]**"

There was a typo in this sentence, we thank the reviewer for pointing it out.

Line 451: "The variability in Larmor frequencies can be reduced by pointing the magnetic field along **[100]** ... "

At [100] the Dresselhaus field does not contribute along the quantization axis, and the variability is attributed to Rashba spin-orbit coupling. At any angle in-between [100] and [110] the spin-orbit variability is a combination of Dresselhaus and Rashba, with Dresselhaus variability being one order of magnitude stronger than Rashba. Therefore, the variability is significantly reduced when the magnetic field points towards the [100] lattice orientation.

We think the statement is clear from Fig. 3a-b and with the available text after the typo is corrected.

26) Lines 353-357: Why is a large Stark shift necessary? Are different Zeeman splittings required for different qubits?

Lines 353-357: "The reduced variability enabled by pointing the field along [100], as in Fig. 3h, also reduces the Stark shift and does not resolve the problem"

As in question 22), our description of this part was limited in the first manuscript. We modified it to describe better these points:

Lines 438-460: “Instead, a more scalable pathway relies on a global microwave field acting on all qubits simultaneously~\cite{hansen2023entangling, hansen_pulse_2021}. Qubits will be driven in resonance with the global field (forming a dressed qubit \cite{seedhouse_quantum_2021}), thus requiring that all qubits have identical frequencies aligned with the resonator frequency. **Individual addressability is then performed electrically, by tuning the qubits via Stark shift in and out of resonance.** However, in the case of a magnetic field along [110], the range of electric control of the g-factors is insufficient to tune them into the same frequency as can be seen in Fig. 3f. The variability in Larmor frequencies can be reduced by pointing the magnetic field along [100] (Fig. 3h) and by reducing the field magnitude to less than ~ 100 mT \cite{zhao_single-spin_2019}. These modifications lead to a standard deviation of less than 200kHz, which is ideal for driving degenerate qubits with high fidelity (assuming a 2MHz Rabi frequency in Ref.~\cite{hansen_pulse_2021}). **The Stark shift tunability also decreases, such that control strategies might need to cope with the inability to tune qubits completely out of resonance**~\cite{hansen_pulse_2021}.”

With respect to the second question, this global control protocol works in a special regime where all Zeeman splittings must be equal. This protocol is different from the most common approach used by most qubit architectures where Zeeman splitting variability is a requirement to address qubits individually. This scenario will be most likely susceptible to frequency crowding and crosstalk for CMOS spin qubits as described in question 22).

27) Fig.4a: What do the black arrows represent?

This is now Fig. 5a

We notice these arrows were confusing so we eliminated them. The black arrows used to represent the displacement of the quantum dots toward each other when the J-gate is switched on.

28) Fig.4d: Which markers are from simulations? Which markers are from experiments? The solid lines are almost indistinguishable.

We have added text inside Fig. 4d, now Fig. 5d, for guidance. The big markers are from experiments. The small points and lines are from simulation. We plotted the lines with different colours to make them easier to distinguish.

29) Fig.4e: Could this be referred to as an inset?

We are now referring it now as an inset inside Fig. 5d.

30) Fig.4h: Are the markers from experiments? Is the histogram from simulations?

Yes, that is right, we have added text in Fig. 4h, now Fig. 5g. for guidance.

“ Histogram of the exchange controllability rates (dJ/dVJ) of device simulations, compared with experimental values (see markers).”

31) Lines 425-427: The authors found that the experimental exchange couplings have a lower baseline than their simulations. Is this in reference to Fig. 4d?

Lines: "We found that the experimental exchange couplings have a lower baseline than our simulations despite a good agreement with the exchange controllability rates $\text{dlog } J/\text{dV}$ (see Fig. 4h)"

Yes, is with respect to Fig. 4d, now Fig. 5d. This figure was modified to include simulations with charge impurities in the presence of interface roughness. Now that the simulated exchange coupling aligns better with the experiment, we rephrased this part to

Line 614: " We found good agreement between exchange simulations and experimental data from devices A and C (Fig. 5d)"

32) Lines 464-470: I agree that the size of the quantum dots have major impacts but do the authors find that the size of the quantum dots have major impacts in this manuscript? If so, where?

We agree with reviewers 1 and 2 that we did not provide enough data to support this conclusion.

We perform new atomistic simulations comparing the dots simulated in the first part of the paper (diameter 15 nm) with larger quantum dots (diameter 18 nm) (See Supplementary Fig 5). The results show that the most concerning part is the larger wavefunction variability and dispersion of the dot centres. Unexpectedly, we didn't observed much variation in valley splittings, g-factors, or valley phases.

We still preserve this conclusion but now we focus on the higher tolerance of the wavefunction of smaller dots to disorder.

Line 667-678: "Secondly, the size of the quantum dots can have a major impact on its performance. The fractal structure of the interface roughness can explain partially this effect, as smaller dots and pitches between dots restrict electrons to regions of the interface with smaller amplitude of the roughness. The average number of trapped charges in the oxide near the dot region also decreases significantly. This leads to higher quantum dot uniformity, and less statistical dispersion in the centres of the wavefunction (see Supplementary Fig. 6). "

33) Fig. A4: How can the interface be made smoother experimentally?

This is indeed a very important question for CMOS quantum technology. We have added a few lines on this conclusion section that refers to Fig.A4 (now Supplementary Fig. 4) regarding the strategies that can be adopted to improve the interface quality.

Lines: 706-714: "Methods to improve the interface quality would involve replicating previously observed correlations between roughness amplitude and different fabrication parameters, such as growth time, oxide thickness, etc \cite{fang_evolution_1997,carim_evolution_1987}. The characterization methods developed in this study can help assess if the improvement in roughness amplitude occurs at the length scales that are more relevant for CMOS quantum processors.

Reviewer #2:

This manuscript reports on the effects of the Si/SiO₂ interface roughness on different spin qubit properties. From experimental TEM images and mathematical reconstruction, interface roughness is computed for different quantum devices. Then, the authors use tight binding simulations (including surface roughness) to compute the quantum dot structures, the excited states (including valley states), and the electron spin g-factor. In turn, a path integral Monte Carlo (derived in another paper) is utilized to study the spin exchange interactions in presence of surface roughness. Finally, the authors conclude that:

- (i) the variability of current devices can be accounted for by the Si/SiO₂ interface roughness
- (ii) a smaller dot size and gate pitches would decrease that variability
- (iii) the electric tuning of the qubits must take into account the surface roughness (dot pulled and displaced along a rough interface).

The authors convincingly study the impact of the Si/SiO₂ interface roughness on the electron spin properties (valley splitting, excited spectrum, g-factor, SOI, spin exchange coupling controllability). Most of the presented simulations (core of this work) are in quantitative agreement with the experimental data.

We thank Reviewer 2 for timely and valuable suggestions. Intrigued by these questions we decided to perform new simulations, which significantly expanded our manuscript and improved its quality. The inclusion of the role of charge traps in one-qubit and two-qubit properties under the presence of interface roughness is the most important addition. Please see our answers point by point.

2) However, the paper let aside the charge traps, which are, to my understanding, the major limitation of the Si/SiO₂ qubit physics and the semiconductor spin qubit in general (Martinez et al. Phys. Rev. Appl 17, 024022, 2022). More precisely, charge traps disturb the electrostatic landscape seen by a spin qubit. For a given (gate) electrostatic configuration, traps will affect the QD location and shape. Therefore, the very first step of prediction/tuning to make an array of quantum dots ready to hold spin qubits is strongly influenced by the presence of charge traps.

We agree that a discussion about the impact of charge traps was necessary. Especially under the light of the paper by Martinez et al.

Figure 1: Device architecture in [Martinez et al. Phys. Rev. Appl 17, 024022, 2022]

The main impact that we observe due to trapped charges in the oxide is to deform the dot potential, which impacts the spin through the roughness of the interface. While the fundamental ingredient still is the roughness, the deformation of the dot impacts how electrons probe such surface, having the potential to create ellipticity, squeezing and in some cases force the dot to move a large distance from the gate centre.

However, we want to point out that these devices differ from ours in at least two fundamental reasons:

1. The spin qubits simulated are driven electrically which requires a high spin-orbit coupling. This can be achieved intrinsically by using hole spin qubits (instead of electrons) or can be induced artificially with the incorporation of micromagnets in the architecture. Both methods make the spin more susceptible to electric disorder, which implies that the qubits have higher variability and are more exposed to charged impurities.
2. The device is formed in a Si nanowire with 3 interfaces surrounding the dot instead of one. Interface disorder caused by Si/SiO₂ roughness and charged traps is present in all of these planes. In planar CMOS devices like ours, there is only one relevant interface.

In the main text we focus on this point:

Lines 107-125 “A more recent paper indicated that a second source - charged impurities in the oxide - could dominate over interface roughness~\cite{martinez_variability_2022}, at least in the case where a micromagnet is integrated to allow electron spin qubits to be driven electrically. Electric driving requires a large spin-orbit coupling, which exposes the spins to the impact of charge impurities. In this paper we focus on spin qubits driven magnetically \cite{Veldhorst2014,vahapoglu_coherent_2022}, which do not have this requirement. These qubits can be controlled coherently without the inclusion of micro-magnets, thus preserving the low spin-orbit coupling of electrons in silicon and protecting the spin from electric fluctuations. We will show in this paper, that under these conditions the remaining spin-orbit variability is interface-induced with charge impurities only affecting the qubit by shifting the quantum dot formation against the roughness profile of the interface.”

Despite these observations, we agree that the role of interface traps can still be concerning for multiple reasons, including its unavoidable impact on two-qubit parameters. So, in this new manuscript we have added the following information:

1. Details in the introduction about different dielectric materials, relevant sources of disorder in Si/SiO₂ devices and the current state-of-the-art knowledge about interface traps.
2. New Fig. 4, comparing simulations with and without interface traps in the presence of Si/SiO₂ roughness.
3. A full section with 4 paragraphs, detailing the role of interface traps for one qubit properties.
4. A whole new version of Fig. 4 (now Fig. 5), in which estimates of exchange coupling are computed in the presence of interface roughness and charged traps.
5. We revised the exchange coupling section, to include the new results.

Once the QD are formed and the spin qubit ready to operate, the typical signature of the presence of charge traps (beyond variability in lever arms) would be a change in the spin exchange coupling compared to a pristine device (which I believe is the origin of the discrepancy between the simulation and the experimental data of Fig.4d).

This insight turned out to be key for reconciling experiments and theory. Initially we had considered other effects that could explain this discrepancy with experimental data, such as valley interference. After reading this question, however, we realized that interface traps could indeed have a dominant role. We have redone the simulations including negative charged impurities into account. The main effect was a decrease of the exchange baseline about 1.5 decades, which agrees significantly better with experiments.

In the revised manuscript, we provide a new version of Fig. 4 , now Fig. 5, with this new inclusion. We also discuss this effect in the text and include previous simulations with just interface roughness in the extended data for comparison.

Another signature will be the qubit charge noise.

Indeed, we have significant electric noise from charge traps, as any dielectric. While reducing charge noise is indeed very relevant for the performance of spin qubits, we are focusing here on the time-independent variability of the qubits properties. That is why we at first disregarded these charge traps. But considering their static field (for slow traps or fixed charges) led to an important impact on variability as mentioned above.

We do a small discussion about these different types of traps briefly in the new section:

Lines 461-474: “Semiconductor devices are in general exposed to the presence of charge impurities in the oxide layer, originated from dangling bonds, Pb-centres, oxide vacancies, and other defects \cite{Müller_2019_traps_solids,elsayed_low_2022}. Some of these are fixed charged traps in the oxide, while others fluctuate over time, which is the origin of 1/f noise in semiconductor devices. The most concerning charges from a variability perspective are charged traps. While two-level fluctuators are still important to understand qubit noise and decoherence, their absolute impact on the variability of qubit parameters occurs at a much smaller scale \cite{stuyck_real-time_2023,zhao_single-spin_2019,ModelingShehata2023}.”

In addition, the work that we have done in this paper can also be useful to investigate how charge noise couples to spin qubits, as effectively, charge fluctuations will cause displacements in the dots in the rough interface. These couple to the dot through Stark shift affecting qubit metrics such as the T₂* [Cifuentes, J. D. et al. [IarXiv.2309.01849](https://arxiv.org/abs/2309.01849) (2023)].

We included this as a conclusion in the new manuscript:

Lines 688-697: “Charge noise coming from two-level fluctuators (TLFs) can also couple to the qubits through a similar mechanism. The induced displacements of the dots in the rough interface lead to small g-factor variations (<1MHz) that can affect important qubit metrics such as the phase coherence T_2^* [cite{stuyck_real-time_2023, Tanttu2019}]. The charge noise couples directly to the qubit Stark shift, so the methods of this paper can be applied to investigate the microscopic nature of this effect [cite{cifuentes2023Crosstalk}].”

When operating a quantum device, a complex sequence including slow/fast changes in either dot potential or spin exchange coupling are necessary. Here, I believe spin properties are computed in a static picture. I think that it would strengthen the paper to discuss these operations (which either modify the dot shapes, and more importantly here their locations) with the consideration of the interface roughness.

As pointed out by the reviewer, qubit operations require a series of voltage pulses to make transitions between initialization, control and readout points. In this paper we are focusing on the static picture, which is already enough to obtain a large amount of information about the physics of the qubits and the role of interface roughness

From this paper it can be concluded that such pulses would involve dot displacements, changes in surface profiles, and so variations in the properties of the dots. This refers to one of the conclusions of the initial manuscript:

Lines: 680-687: “Another conclusion is that electric tuning of qubit frequencies (using spin-orbit effect) and exchange coupling (using barrier gates) both rely to a large extent on moving the quantum dot laterally, dragging it against the rough interface. This may lead to considerations in future designs of quantum dots and the methods for characterising the interface”

Now, the next steps would involve revising the time-dependent evolution of the spin-state during these operations. While this is an important and extremely interesting project, we think that doing that work with the level of detail that it requires (including sources of noise, ramp times, etc) would highly exceed the scope of this paper. We also consider that to do that it is necessary to have the static picture completely clear, which is the role of this paper.

Are the simulations performed for a single electron per dot, or 3? The question stand for all simulations when compared to the experiment. Am I right saying that the 1,3 or 3,1 charge states were used to enhance the exchange coupling between the 2 qubits?

The (1,3) electron configuration brings a number of benefits, including a higher exchange coupling, which affects the way the simulations are compared with experiments. This is something we should have mentioned in the manuscript.

We added the following text.

Lines 594-603: “Our dataset combines qubits at the outer shell of quantum dots with one electron and 3 electrons. The exchange interaction is larger for higher electron numbers, due to the increasing overlap between the wavefunction of the quantum dots. This is observed in device A, where the exchange baseline of the (3,1) configuration is higher than in (1,1). These situations are so far not considered in the simulations, which are performed in the (1,1) electron configuration.”

I find the concluding comment “smaller dots and pitches between dots restrict electrons to regions of the interface with smaller amplitude of the roughness, reducing the effects of valley interference” hard to understand. On one hand, closer spatial positions are more likely to have the “same z”, and therefore the phase difference in the valley state of 2 adjacent qubits are less likely to differ. On the other hand, bigger dots average out the roughness (to a certain extent), in the same fashion as for motional narrowing. A discussion on the valley phase difference as a function of the dot size and the gate pitch would help the reader understanding the authors’ conclusion.

We acknowledge that we did not provide enough justification about this in the main paper. As reviewer 1 made a similar question (Question 32), we decided to perform new simulations to prove this point.

We simulated larger dots atomistically in new Supplementary Fig. 6. However, in the results we did not observed major variations in the valley splitting, nor in the valley phase. We did observed changes in the wavefunction that are very relevant and tend to signal that smaller dots are less susceptible to disorder. A new conclusion was included in (Question 32, Reviewer 1) based on these arguments.

We took out of this conclusion any aspect relevant to the valley phase as we consider that making this claim requires an in-depth understanding on how the valley phase varies with the interdot distance, and how likely is a valley phase shift to occur when the quantum dots are displacing towards the centre under the action of the J-gate.

I find the claim of lines 65-67 on the improvement of the fabrication processes letting the interface roughness the only property that is intrinsically limited very strong, especially from the perspective of charge traps (which could be discussed in terms of percolation density or charge noise).

We have rephrased this part to add the new results in the light of the new simulations.

Lines 659-667: “The main conclusion is that most of the qubit variability in current devices is explained by the roughness of the Si/SiO₂ interface roughness. The presence of charge impurities is also significant regarding two-qubit properties. Other effects (such as strain inhomogeneity and geometrical deformation of the gates) can be mitigated down to levels that are, at most, comparable with these intrinsic mechanisms.”

I also have a list of minor comments, which could be addressed to clarify the discussion:

- Line 748, I believe [110] should be [100]

That is correct, it should be [100]. We made the correction as suggested.

- Fig. 4g, I don’t understand to what correspond the green line

We have replaced the green line in Fig. 4d and Fig.4g , now Fig. 5d and 5f , by a shaded region (yellow color). The text “Experimental range” is included in the figure. This represents the relevant scale where exchange interactions can be measured in an experiment.

- The definition of k_0 (given line 666) could also be written in the caption of Fig.2

We added this information in the caption of Fig 2e.

“ The oscillation wavelength is $\frac{4\pi}{k_0}$, where $k_0 = 0.82 \frac{2\pi}{a_0}$ is wavevector of the conduction band minima in the silicon crystal.”

To conclude, I think that this study, however convincing for the interface roughness, does not yet fit the wide scientific readership of Nature Communication. The discussion must include a part on the charge trap aspect of the materials considered, either based on experimental work or simulations. If additional work on that aspect cannot be done, the scope of the paper must be narrower, or could include a discussion on the different operating regime of a quantum dot array (strong coupling for 2-qubit gates, weak for idle and single qubit, etc.) and the impact of interface roughness on the qubit when “operating” an array.

We agree with the reviewer and decided to expand the work to include the treatment of charge traps. The reviewer was correct in their predictions and we thank them for the significant improvement to our manuscript that this addition represented.

We understand that this addresses the reviewer main concern for publication in Nature Communications.

Reviewer #3 :

The manuscript "Bounds to electron spin qubit variability for scalable CMOS architectures" gives a detailed discussion on the influence on the Si/SiO₂ interface on the operation of spin qubits formed at this interface. Utilizing simulations and measurements the work presented assesses the viability for scaling of spin qubits formed with this architecture.

The work is timely and of high interest

We thank the reviewer for considering our work timely and of high interest.

Unfortunately the manuscript in its current state has, in my opinion, two overarching issues. First, the authors fail to contextualize their work within the wider field of CMOS compatible spin qubits.

We agree with the referee that adding some context would be positive for the general readership of Nature Communications. Our original manuscript was written explicitly refraining from extrapolating conclusions and comparison with other CMOS geometries and material stacks due to the lack of comparable studies with large number of realistic simulations and devices (with few potential exceptions, such as the work from the Niquet group). But in retrospect, it is still important to discuss the general issue of CMOS compatibility and the particular relevance of the Si/SiO₂ interface. This corroborates the argument that these findings are transferrable and technologically relevant.

Second, I struggle to find explanations for a number of critical technical details in the manuscript and the references therein. I will go through the manuscript and highlight the specific paragraphs that I either cannot comprehend or disagree with.

Please see our answers on each of these matters point by point.

In the introduction the authors state that "Enabling the breakthrough applications of quantum computation requires millions of qubits to perform error correction [7, 8], and it is infeasible to address all of these qubits with individualised pulses catering to their variable parameters." While this is a notion generally agreed upon in the field, the authors then got to simulate an array of 49 (7x7) qubits and compare to measurements on 12 qubits from 6 dual qubit devices. This raises the first technical questions: Why do the authors choose to simulate a 7x7 array rather than a set of 2 qubit devices in the same/a similar configuration as the fabricated devices? I cannot understand from the manuscript why the simulation of an array is beneficial in any way.

This question relates to an omission in our part – a proper discussion of the role of self-similarity in the interface disorder. We thank the reviewer for bringing this matter to our attention.

Firstly, we acknowledge that the reviewer is correct when they mention that most of the results would be captured by a simpler 2 dot model. Indeed, such model allows us to calculate all relevant properties for gates between spin qubits, and all the experimental data was obtained under this reduced model.

However, the disorder in a larger array, such as the 7x7 array modelled here, is not mutually independent between all the pairs of dots. Since the interface is self-similar, the amplitude of the roughness is larger across such an array than it is at smaller scales.

In reality, the simulation consists of a single computer-generated interface large enough to accommodate the whole grid, but each qubit or pair of qubits is then simulated from an atomistic point of view in simulation cells containing one or two dots only, with its corresponding patch of the total interface.

These points were clarified in the revised manuscript with an early mention of this fact at the introduction.

Lines 126-138: “[...] This work combines measurements of 12 qubits across 6 different CMOS quantum dot devices, transmission electron microscopy (TEM) images of cross-sectional cuts of 6 other quantum dot devices and theoretical analysis of quantum properties of electrons in simulated quantum arrays of 49 quantum dots. **Such a geometry allows us to study the impact of the self-affine scaling of the Si/SiO₂ roughness on qubit properties at different length-scales.**”

We mention it later in the Si/SiO₂ roughness section

Lines 149-155: “The Si/SiO₂ interface has an intrinsic fractal structure \cite{Yoshinobu1995}, that has not been considered in previous variability studies. Our decision to simulate a 7 \times 7 dot array under the global rough surface (Fig. 1b) is motivated by understanding how this fractal scaling of the interface roughness impacts qubit properties at different length scales. [...]”

Another issue that arises is that 49/12 qubits are nowhere near a few million. Note that a single failure in the simulations/measurements corresponds to 10.000s/100.000s of devices failing in a processor that contains a few million qubits.

We agree with the reviewer. The variability of electron spin qubits does lead to important questions about the number of out-of-range qubits one would have in the context of control pulses that are not tailored for the particularities of each qubit. Our manuscript provides estimates on the ability for such quantum dots to electrically tune the qubit properties to align with the allowable range for high fidelity operation, but a significant number of qubits would be defective in a million-qubit array if the current estimates remain accurate for future devices.

We would highlight that we identify this as a strength of our manuscript. Architectural designs of quantum computing systems only make sense in the context of a realistic view of yield and statistical dispersion of qubit properties.

Other qubit technologies also face this challenge and a realistic analysis of the yield of qubits with workable parameters is key for a sensible approach to scaled up quantum processors.

We mention that a number of mitigating strategies are available to minimize this impact. Many groups, including ours, already adopt multielectron spin qubits, such that defective quantum dots that present off-range single electron parameters may be recovered by, for instance, adding two more electrons to form a closed shell state. Moreover, research into the materials and fabrication processes may improve the qubit uniformity. Combining these solutions with a software-level consideration of these defective sites is expected lead to a workable architecture in the future.

The authors then go on to state "Variability may be caused by a few factors, such as strain, fabrication defects, accidental introduction of charged impurities in the oxide and so on [11–13]. Ultimately, most of these sources of variability can be improved with increasingly precise fabrication – industrial foundries focus most of their efforts in addressing these issues [6]. The only source of qubit variability that is inextricable to CMOS technology is the roughness of the interface between

the crystalline Si and the amorphous SiO₂ ..." Reference 6 and 12 are papers discussing properties of CMOS spin qubits, reference 13 is a paper simulating the impact of strain and dopants on CMOS transistors, and reference 11 is a review discussing material requirements for Si based spin qubits. Unless the argument is that CMOS spin qubits have been manufactured and hence foundries are doing a good enough job of controlling the process parameters, I do not see how these reference justify the authors statement that "most" variabilities can be improved by fabrication but the oxide interface is bound to stay as is.

We understood the reviewer's point and agree that as much as there is progress in all these fronts, there will never be a scaled-up device that has completely removed all of these effects. This statement was phrased by us in a simplistic way, mostly to reduce text length. In view of the reviewer's criticism, we significantly expanded our considerations to include a treatment of charged impurities.

A thorough analysis of the results from this simulation campaign can be seen in the response to Reviewer #2.

We have performed strain simulations and found those to have much less impact on qubit parameters. Fabrication defects may, of course, be dramatic enough that no qubit is formed, but the more typical problems with small misalignments and non-conformal gate geometries play a small role that is easily compensated by adjust DC voltage biases. Charged impurities and two-level fluctuators are, however, a far more concerning problem indeed. We discuss this next.

While it is true that CMOS manufacturing has made impressive strides, issues like the control of vacancies [<https://doi.org/10.1109/JPROC.2012.2189786>] and charges trapped on the oxide [<https://doi.org/10.1109/TED.2019.2907816>] have been issues for decades and may very well influence qubit variability.

We agree that this is the case. A more accurate statement is the following: in an idealised theoretical model where charge traps are the only issue and interface roughness is absent, charge traps would not impact qubit variability. Alternatively, one might state that the main impact of charge traps is to impact the electrostatic quantum dot formation, which impacts the qubit through the exposition of the spin to a different Si/SiO₂ interface environment. We have added an entire section about this in the new manuscript (and Fig. 4) talking about the role of charged traps on one-qubit properties. We also included charged traps in two-qubit simulations, which turned out into a better agreement with experimental data.

Furthermore, while the Si/SiO₂ interface is and always has been prevalent in CMOS devices, other oxides/materials have been evaluated [[10.1088/1674-4926/38/7/071002](https://doi.org/10.1088/1674-4926/38/7/071002), <https://www.mdpi.com/1996-1944/5/8/1413>] and to the best of my knowledge implemented for the production of memory devices [<https://www.mdpi.com/1996-1944/3/11/4950>, <https://doi.org/10.3390/nano13172456>].

Due to the significantly higher density of interface traps in Si/high-k interfaces, almost without exception such materials are deposited on top of a thin thermal SiO₂ layer. This means that the roughness of the interface to which the spin is subject is still due to the chemistry of Si/SiO₂ (although perhaps with different roughness amplitudes, etc).

We have rewritten this paragraph to include more details about semiconductor qubit devices, architectures and the main sources of variability that we are concerned about

Lines 59-106: "Qubit variability is caused by the same factors that affect conventional CMOS technology, such as strain, fabrication defects, accidental introduction of charged impurities in the oxide, interface roughness, and so on \cite{asenov_simulation_2009, Cheng2009}. Industrial foundries focus most of their efforts in addressing these issues \cite{zwerwer_qubits_2022, elsayed_low_2022}, with a central role played by the choice of materials for substrate, dielectrics, gate metals, etc \cite{Saraiva2022}.

In the centre of the discussion is the dielectric interface where the electrons are confined \cite{Lawrie2020, Burkard2023_semiconductor_qubits}. High fidelity silicon qubits have been measured in Si/SiGe and Si/SiO₂ heterostructures \cite{philips_universal_2022, mills_two-qubit_2022, noiri_fast_2022, Yang2019, tanttu2023}. In the case of Si/SiGe heterostructures, a thin layer of uniaxially strained silicon binds the electron due to the conduction band shift caused by the strain when compared to the relaxed Si_xGe_{1-x} alloy. The Si/SiGe interface provides, in general, reduced levels of interfacial disorder compared to that of Si/SiO₂. Potential shortcomings of Si/SiGe technology are the reduced gate control when compared to MOS devices and the limited tolerance of the material stack to high temperature annealing processes, commonly adopted in the CMOS industry. More information on this material and its comparison to oxides can be found in Ref. \cite{Saraiva2022}. In the context of spin qubit variability, comparing an oxide interface to Si/SiGe is hard because the nature of disorder in the two materials is different - alloy disorder and miscut angles in SiGe, compared to amorphous oxidation in SiMOS. Moreover, the dominant effect of spin-orbit coupling being studied here is masked by the presence of micromagnets which are commonly adopted in SiGe qubit architectures \cite{philips_universal_2022}.

In this paper, we focus on qubits at the Si/SiO₂ interface. SiO₂ does not have a regular lattice structure when thermally grown on the silicon surface, so the interface is atomically rough. The higher levels of interfacial disorder when compared to Si/SiGe are attributed to this roughness and to the presence of fixed charge defects that can be either at the Si/SiO₂ interface, in the bulk of SiO₂ or at the metal-oxide interface \cite{elsayed_low_2022, Intel2019, Müller_2019_traps_solids}. Potential advantages are in the higher electrical tunability and compatibility with conventional CMOS technology, which benefits the integration with on-chip electronics \cite{Veldhorst2017}."

Most importantly however, the authors completely ignore that CMOS compatible spin qubits have been successfully implemented using a buffer layer between the oxide and the charge carrier forming the spin qubit both in Germanium and Silicon in order to mitigate the issues arising from the oxide interface [<https://doi.org/10.1002/adfm.201807613>, <https://doi.org/10.1038/s41586-022-05117-x>, <https://doi.org/10.1063/5.0002013>]. Evaluating whether the introduction of the buffer layer is advisable could have been one of the major contributions of this work. Instead, other CMOS compatible qubit architectures are almost completely ignored by the authors. When addressing a wide, multidisciplinary audience as in Nature Communications this seems very careless.

The reviewer is referring here to one of the leading material stacks for qubit implementation, a uniaxially strained quantum well of Silicon formed between two layers of SiGe alloy. We agree with the referee that the readership would benefit from a more explicit discussion of this material given its significant progress, and we have done so in the revised paper.

We point out, however, that at this moment there is not enough information about the intrinsic sources of variability of spin qubits in SiGe, so it is hard to draw this comparison appropriately. This is the reason why we refrained from discussing SiGe in the first version of the manuscript.

It is then discussed how a 3D model of the Si/SiO₂ interface for tight binding simulations is built from the 2D TEM images. As already mentioned before it is not clear to me why the authors chose to simulate a 7x7 array.

This is explained above when the reviewer first mentioned this issue.

More importantly though, the authors then go on to model the amorphous silicon as a "virtual crystal approximation" using "SiO₂ atoms" on a silicon crystal lattice making the argument that this is standard in the field based on a single 10-year-old reference. This seems ill-advised. Real amorphous materials show variability in atomic configuration [[https://doi.org/10.1016/0022-3093\(76\)90023-5](https://doi.org/10.1016/0022-3093(76)90023-5)] that might very well influence both the local strain and charge density under the interface. Effectively ignoring the amorphous nature of the material when explicitly evaluating the interface between an amorphous and a crystalline solid seems fundamentally flawed. It certainly requires more justification than a single 10-year-old reference hidden in the Methods section. For example, a clear mention of this circumstance in the main manuscript and a paragraph discussing the potential shortcomings of this approach are in my opinion absolutely necessary.

The reviewer is correct in their criticism. It is, indeed, a leap of faith to trust that the virtual crystal approximation (VCA) would capture the physical complexity of an amorphous oxide.

We have now reviewed our text to be upfront about this shortcoming of the simulation.

For the reviewer's interest, we point out that we do not believe any of our current methods would be capable of accurately describing the complexities of the chemistry of SiO₂ while simultaneously being capable of simulating the millions of atoms contained in a quantum dot. Hence the need for this approximation.

As far as our knowledge goes, this problem would be pervasive to all electronic structure methods, given the complexity of simulating the charge transfer processes incurred in the oxidation and reduction of silicon atoms in combination with the kinetics of oxidation under various dry and wet oxide formation processes. VCA is our best guess on a method to incorporate elements of the roughness of the interface without complicating simulations to the point that we wouldn't be able to have a sufficiently large simulation cell. We added another citation in the methods section about this topic [Rahman, R. *et al.* Phys. Rev. B 83, 195323 (2011)]

The paper that the referee cited nicely illustrates the complexities of amorphous SiO₂, but it only simulates 336 atoms of SiO₂ from Molecular Dynamics. This is very far from providing all the answers we need, which need to include a model of the interface between Si and SiO₂, external fields, spin physics and an accurate solution of the electronic wavefunction. In addition, the simulation of a single instance of SiO₂ is not the end of story as various other permutations and configurations of the oxide are possible depending on the fine details of the oxide growth parameters.

In the new text we mention the Virtual Crystal approximation in the main manuscript.

"Despite SiO₂ not having a regular lattice structure, we simulate it by assuming an atomistically ordered virtual crystal approximation (see Methods). The material is endowed with the same lattice structure as Si and the tight-binding parameters are set to emulate the electronic structure of the interface~\cite{kim_full_2011,Bersch2008}. This approximation allows us to simulate interface disorder atomistically as seen in Fig. 2a."

While probably not a scientifically accurate way to confirm the validity of the method, there is encouraging agreement with experiments when we adopt this model. We can confidently say that

our work has improved very significantly the realism of simulations, which is why we have obtained the most accurate description of the intrinsic spin-orbit coupling of CMOS spin qubits achieved so far.

Starting the comparison between the simulations and measured qubits, it is stated that "all devices with functional gate electrodes (as determined by their influence on the charge sensing single electron transistor) could form controllable pairs of dots." However, the yield is not mentioned. For the validity of the claim it is very relevant where 1% of all checked devices have "functional gate electrodes" or 99%.

The current yield per gate of our process in a University cleanroom is 95%. The yield per gate in a foundry fabrication process would be hardly distinguishable from 100%, yielding hundreds of billions of gates per chip.

In this section two things are furthermore unclear to me:

1) I do not understand what is shown in Fig. 2e

It is the Bloch oscillations in the z-axis caused by the presence of valley states in Si. There is a section in the methods section about this. We have added a reference to this for clarity

Caption: Visualization of the valley oscillations parallel to the $[001]$ lattice orientation. (see **Methods section**)

Citation in the text: "The electron density will present Bloch oscillations in the z direction, which are barely visible in Fig. 1f and were enhanced in Fig. 2e by taking the difference between the electron densities of both valley states (**Methods**)."

2) In Fig. 2f, g the simulations are compared to 2 devices. Why only 2 and not all 6?

An accurate determination of valley splitting requires measuring the spin relaxation of qubits as a function of magnetic field and gate voltage. Since this relaxation time is very long (typically in the 10s to 100s of milliseconds) and ramping the magnetic field can be very slow (it takes several minutes for the superconducting magnet to stabilise again), these experiments are very time consuming.

Because of that we do not measure the valley splitting in all devices. In the vast majority of the cases the valley splitting in our quantum dots is large enough that we can go straight to measuring the spin qubit properties.

It is then stated that all 49 qubits in the simulation can be tuned to a valley splitting larger than the Zeeman splitting. However, this generic statement while crucially important, is not backed up with any further insights. Given that these are simulation results why do the authors not explicitly state/plot what the distribution of the valley splitting in the 49 quantum dots looks like?

We have now included the full distribution in the main text, on the side Fig 2f.

In principle it should be possible to evaluate the expected (best case) fidelity of the quantum bits from the calculations? Why is this not done? Given the information currently in the manuscript the statement that "a small but finite number of quantum dots that have valley splittings clashing with the spin splitting" which can be "discarded on the firmware level" seems unjustified. The authors provide neither the necessary statistics nor insights to substantiate such a claim.

The precise numbers for the infidelity caused by spin variability will depend on the particular details of how the qubits are operated and how well the microwave driving is performing. We can provide an example of such analysis in a recent paper by Hansen et al. to demonstrate how complex such analysis becomes. This level of detail would of course more than double the length of the current paper and is, therefore, considered out of scope.

We agree with the referee, however, that a statistical analysis is due if we are to claim the number of off-range qubits is small enough that it can be treated at the firmware level. In our 49-qubit simulation, only one of the qubits is suspiciously departed from the average and lacks controllability to the point that it would probably need to be discarded. We mention this now in the main text.

We do not have, however, the means to gather enough statistics to quantify the percentage of the qubits that would fall in such category. If this number is smaller than 2% as this simulation would seem to indicate, we would need approximately 500 simulations to pinpoint the exact percentage with enough precision. We do not have sufficient computational capabilities for such an undertaking.

For the referee's appreciation, the current manuscript contains results from approximately 1 million cpu*hours.

In Figure 3 the authors evaluate the variability of the spin-orbit coupling of the different qubits. First, it is unclear what the magnetic field used for the simulations here (Fig.3 a) is (I assume still 700mT?).

The g-factor is independent of the magnetic field. We describe it in units of [MHz/T] in the paper.

In this case, we used a magnetic field of 1T for atomistic simulations, but the results are independent of this value, which we confirmed separately.

We added the following lines in the methods section for G matrix computation:

Lines 948-957: "For all simulations, we set the magnetic field magnitude amplitude to 1T and vary only the field orientation \hat{B} . The spin g-factor is independent of the magnitude of the magnetic field as long as the valley splitting is non-degenerate with the Zeeman splitting [Bourdet2018,gilbert_-demand_2023]."

Second, I cannot find an explanation for g_1 and g_2 and hence don't understand what frequency difference g_1-g_2 refers to.

Each device was measured in a double dot system as described in the introduction. We are just showing the difference between the g-factors of the two qubits on each device (g_1 for the qubit below P1, and g_2 for qubit below P2). We made the following modifications on the caption to clarify this point.

"**a-b**, Comparison between g-factor variability in atomistic simulations and measurements in devices A to E under a varying magnetic field angle. We compare in **a** the difference between the Larmor frequencies of the two-qubit qubits measured in each device (g_1-g_2) μ vs the difference between the frequency of neighbouring dots (Fig. 1a)."

Finally, it is stated that "All 12 qubits measured in devices A to E show behaviours consistent with this description." First, - A, B, C, D, E - are only 5 devices and hence only 10 qubits and not 12 or am I missing something here?

In device A we provide data from spin qubits at (1,1), (3,1) and (1,3) configurations.

Qubits formed at the outer shell of quantum dots with different electron numbers have very different g -factors, even if they are under the same gate. This is most likely due to substantial differences in surface profile of single electron dots and the density function of 3 electron dots, as well as the difference in exposition to the atomic profile of the interface for electrons in different valley states.

We have included the following text in

Lines 370-376: “Qubits in the same quantum dot but with different electron numbers are considered as different in the total count, as they have substantially different g -factors (see data from device A in Fig. 3b). This is most likely due to different exposition to the atomic profile of the interface for electrons in different valley states.”

Second, for 3 of the 5 devices only one measurement point seems to be shown, which is consistent with almost any description.

These devices were measured in a regular single axis magnet, which only provides magnetic fields for a fixed angle. Vector magnets are much more costly, so we do not have one in every measurement setup. Notice that Table 1 specifies which experimental setups include a vector magnet.

We point out, however, that the consistency of these data with the description is not to be taken for granted. For instance, matching the experimental data in holes with such analysis would have been significantly more difficult. There is not particular reason why the sinusoidal dependence of the spin-orbit effect with the external magnetic field has to be centred at that particular value of g . This is a result of the significant difference between Dresselhaus and Rashba effects in Si/SiO₂, a somewhat counterintuitive effect (Dresselhaus in bulk silicon is zero by symmetry) that is particularly well explained by our theoretical analysis.

It is then mentioned that a few simulations as well as one device (in a particular configuration) escape the overall trend and simply stated the this can be tuned away. However, in the introduction the authors explicitly state that "it is infeasible to address all of these qubits with individualised pulses catering to their variable parameters". So how is it feasible to tune the valley splitting of individual qubits?

We can tune the valley splitting with a DC offset bias. While individualised pulsing would be hard, individualised DC biases can be efficiently provided in a number of ways. For example, a charge holding capacitor can be connected to the gate if fine tuning of the voltage is needed. For the kind of coarser tuning needed to address the valley splitting, a simple floating gate architecture can hold the charge needed to set the DC bias to a desirable level.

We have rephrased the part mentioned by the reviewer in the main text:

Lines 41-50: “And it is infeasible to address all of these qubits with pulses catering to their particular parameters with wires individually running from the room temperature controllers. Instead, this variability must be embraced and corrected through the combination of on-chip electronics operating at cryogenic temperatures^{\cite{Veldhorst2017, vandersypen_interfacing_2017}}, and robust quantum control pulses that will be shared among several qubits^{\cite{hansen_pulse_2021,hansen2023entangling}}”

And if it is not, what are the implications for manufacturing millions of qubits? In addition, the authors find the "electric control of the g -factors is insufficient to tune them into the exact same

frequency" and then conclude "Therefore, strategies for qubit control need to be designed to circumvent this variability and tolerate the natural dispersion in qubit frequencies introduced by the oxide interface." What strategies are the authors alluding to here?

Please note we have rephrase this part (lines 438-460) due to related questions from reviewer 1 (Question 22 & 26) .

These strategies would have to combine solutions to 1) reduce the surface roughness with better oxidation processes; 2) minimize the impact of g-factor variability by developing low field control methods and 3) develop robust gate pulses that can cope with variability of qubit parameters.

These are discussed in Hansen et al., so we refer now to these papers in this part of the text. [Hansen, I. et al. Phys. Rev. A 104, 062415 (2021), Hansen, I. et al. arXiv.2311.09567 (2023).]

As best I can tell, one could also conclude that device architectures that avoid forming qubits at rough interfaces are more promising candidates for realizing spin qubits. As mentioned earlier, the authors consistently ignore alternative device architectures for no apparent reason.

We are unaware of any silicon interfaces devoid of roughness.

If the reviewer is referring to Si/SiGe, that interface is intrinsically rough due to alloying effects and miscut angles (resulting from the crosshatch patterns stemming from the dislocations that are generated in the graded buffer region of the SiGe substrate). That is one of the reasons why MOS quantum dots yield consistently higher valley splittings than SiGe quantum dots.

Unfortunately, no studies have been performed thus far on the full range of qubit variability to be expected from SiGe, so it is impossible for us to compare the two technologies in a scientific manner. Hence the absence of commentary on that platform in our initial submission.

In the revised manuscript, however, some general comments on that platform are added to make sure that any readers unfamiliar with the field can appreciate that other materials might have other variability profiles.

In the final section the interaction between neighboring qubits is investigated. A path integral method is used. Again, a number of technical question arise that I cannot find an explanation for in the manuscript. Based on Figure 4c it looks like to electrons are moving in a parabolic potential. How is this potential generated? What justifies its symmetries?

Potential configurations for each value of J obtained from finite element simulations in COMSOL MULTIPHYSICS (See Extended Data Figure A2 and Methods section for Modelling of digital twin of the devices and electrostatic simulations). This is a realistic potential simulation based on the methods described in the paper, we do NOT assume any symmetry.

We have included this reference in the caption of Fig. 4e, now inset of Fig. 5d, for clarity:

"This potential is obtained from finite element simulations in COMSOL MULTIPHYSICS (see Supplementary Fig. 2 and Methods"

How is the amorphous nature of the oxide implemented? It is stated that the "Interface roughness can be readily included by defining a 3.1 eV step potential barrier to simulate the conduction band offset between silicon and the SiO₂ layer." I do not understand how the authors "readily" move from

a physical roughness on the atomic scale to a potential barrier which almost certainly won't be atomically sharp due to the quantum nature of the electron wave function in the material stack.

As discussed earlier, the simulations presented here are not compatible with an appropriate description of the chemistry of oxi-reduction due to the size of the simulation cell. Full detail of the model is presented in the Methods section.

The simulations are compared to 2 devices again. Why not all 12? How are the devices the authors choose for comparison selected?

For clarity, we have 12 qubit configurations in total in this study, which means we would have 6 pairs of qubits to analyse.

These 6 pairs used different gate and dielectric stacks. Only devices A and C are nominally identical. In the case of exchange coupling controllability, these differences impact the final shape of the dots significantly enough that it becomes hard to compare.

We added further focus on this in

Lines 603-613: " The gate stack also has a significant role in the exchange control, affecting the effectiveness of the electric fields generated by the J gate in the channel. Here we compare devices A and C, which are both made with Pd/Ti gates with ALD oxides (see Table \ref{tab:Devices} and device architecture in Fig. 1f. Exchange control was also measured in devices E and F with Al gates in Ref.~\cite{tanttu2023}, observing larger control rates and variability due to the absence of the ALD oxide and irregularities in the gate structure \cite{Saraiva2022}."

The results are that "the experimental exchange couplings have a lower baseline than our simulations despite a good agreement with the exchange controllability rates". This disagreement is attributed to "valley interference in the devices". Again, then why are these 2 devices chosen for comparison if they are suspected to have valley interference? It was stated earlier that devices can readily be tuned out of valley interference?

There is a conceptual difference between valley degeneracy and valley phase interference.

A valley degeneracy occurs when the valley splitting of a single qubit is more or less the size of the Zeeman splitting. That creates some ambiguous spin-orbit coupling as we explain in the text. That effect happens at a small voltage region, so the effect is tunable [Gilbert, W. et al. Nat. Nanotechnology 1–6 (2023) doi:10.1038/s41565-022-01280-4].

Very different from this is valley phase interference. Any qubit has a different valley phase. That's the figure we plotted in Fig. 3e. Interference between the valley phase of two neighbouring qubits can affect the exchange interaction between them. Essentially all qubits will have some degree of valley interference, unless they have fortuitously identical phases.

Did the authors attempt this? Would it also be feasible to conclude that the approximations made in the simulations might cause the disagreement between measurement and simulation?

That is precisely what we conclude. All devices will have some level of valley interference. Because our model does not capture that, all of our estimates of exchange are overestimated.

We note, however, that due to the valuable suggestions from reviewer 2 we have now realised that part of the disagreement could be attributed to fixed charges in the oxide, which are known to be present in all CMOS devices.

The section concludes with the statement "In the specific geometry simulated and measured here, the tunability ranges from 6 to 10 decades per volt, large enough to compensate for the interface disorder and consistently hit a target "on" exchange rate across all devices." Does this mean all simulated or all measured devices? Given that this is a fairly important statement - where is this demonstrated or why is it not demonstrated?

It is demonstrated in Figure 5d, which shows that the range of desirable exchange couplings (from 100kHz to 100MHz) can be achieved by biasing gates within a range of approximately 0.5V.

This is observed in 100% of the devices where we attempted to measure exchange (excluding, of course, the ones with broken gates, etc) and 100% of the simulations.

In the conclusions the authors argue that "Finally, this study realistically sets the ultimate variability of qubit parameters." I do not understand this statement. The study investigated variability of qubit parameters of one isolated issue (Si/SiO₂ interface roughness) in a specific qubit architecture. It seems absurd to claim that this sets any ultimate limits on qubit parameters in general.

This was an inadequate choice of words. We have included that this is the

Line 725: "ultimate variability of spin qubit parameters in CMOS devices".

In addition, the authors state "These results outline the minimum demands for an architecture that can deal with qubit variations while maintaining high fidelity." Similar to the previous statement this seems completely unjustified. Nothing in this work addresses qubits in general and as mentioned several times before, even when looking at CMOS compatible electron spin qubits in silicon, architectures that avoid the main/only source of variability investigated here are readily available and have been demonstrated experimentally.

We have changed the choice of words to

Line 744: "minimum demands for a CMOS spin qubit architecture."

CMOS architectures refer to metal-oxide-semiconductor, which is precisely what is studied here. What the reviewer considers "CMOS-compatible" is unclear from their comments. We believe we addressed the whole gamut of spin qubit quantum dot materials in our responses above.

In my opinion the manuscript is not publishable in its current state. The following issues would have to be addressed:

First, a number of overly general statements need to be adjusted.

Hopefully we addressed this concern with the extensive revisions of our manuscript.

Second, the choice of the specific qubits (chosen from the 12 available) used for comparisons to the respective simulations needs to be justified.

We hope to have clarified this aspect as well. Most importantly, we can state clearly that we have not "cherry picked" the results that match well our simulations. Any devices left out of comparisons were either not measured for that particular data (for practical reasons) or are not comparable due to details of variations in materials, etc.

Third, the missing technical details outlined above should be addressed. In particular, approximations made in the respective simulations should be mentioned and their impact on the results should be discussed. Furthermore, all (free) parameters used in all simulations need to either be stated or sourced so that the work is reproducible.

We strongly agree with the reviewer and tried to be as thorough as possible in our revisions addressing this point.

Fourth, other CMOS compatible spin qubit architectures need to be mentioned and the authors need to justify the choice of the architecture investigated here.

We agree that mentioning other architectures is appropriate. We also agree that our choice of architecture needs to be well motivated, which we firmly believe is the case. See point below.

After addressing these points, the manuscript would in my opinion be publishable in a specialized journal. In order to justify publication in a multi-disciplinary journal with a large audience like Nature Communications, the authors would have to furthermore show that their work has an impact on semiconductor spin qubit devices beyond the specific device architecture investigated in this work – i.e. answer the question why the impact of the Si/SiO₂ interface on qubit operation cannot simply be bypassed by either the introduction of buffer layers or the use of other CMOS compatible (potentially crystalline or poly-crystalline) oxides.

We refute the reviewer opinion that to justify publication in Nature Communications, the work has to impact semiconductor spin qubit devices beyond Si/SiO₂. The Si/SiO₂ interface is at the heart of a multi-trillion-dollar industry and has shaped our era. Addressing how this interface affects spin qubits is of general interest to all scientists in our opinion.

We also hope to have made clear that the concept of a “buffer layer” that resolves the issue of interface roughness and qubit variability is not compatible with the most current understanding of the material science behind silicon spin quantum computing. Variability of qubits, large or small, is a key architectural design parameter that needs to be considered in all technologies.

We include the following lines in the conclusion:

Lines 716-723: “This work focused on the case of Si/SiO₂ interfaces, which has been studied enough that we are able to draw firm conclusions on its impact on qubits. Other oxides or dielectrics might also be understood adapting the methods developed here, providing pathways to shortcut the qualification of material stacks for quantum processor fabrication with the assistance of theoretical calculations.”

REVIEWER COMMENTS

Reviewer #1 (Remarks to the Author):

The authors have addressed some questions, but perhaps not in full depth. I believe that preparing the manuscript with greater attention to detail would enhance its appeal. Below, I have outlined my questions corresponding to specific points.

3) Fig.1a: It is unclear how to align all the electrodes (P gates and J gates) for 49 qubits. Are SETs still used for read-out at this stage?

We have not explicitly committed to any one form of gate layout for this work to avoid confusion between metal gate stack architecture and the role of the interface. An example showing methods to implement such square grid of quantum dots is discussed in Veldhorst, M. et al Nature Communications 8, 1–8 (2017).

In the manuscript, the authors mention a 49-qubit system, which naturally leads readers to wonder about the specific configuration of the gates for these qubits. It's crucial to consider that if the gate arrangement differs from what is described in your paper, it might not be straightforward to extrapolate your results directly to a 49-qubit system. Therefore, it would be beneficial to discuss or address this aspect somewhere in your work to clarify any potential discrepancies or assumptions regarding gate layouts for larger qubit arrays.

5) Fig. 1h: Why is the correlation length 500 nm for 49 dots, 100 nm for 4 dots, and 10nm for 1 dot?

We have included this text in the caption of Fig 1h for clarity:

“The data is plotted as a function of $\lambda=2\pi/q$, where q is the wavenumber. This allows us to compare λ with most relevant length scales, namely is the silicon lattice parameter $a_{Si} = 0.357$ nm, the dot radius of 10-15 nm, the double dot size of 80-100 nm and the full 500 nm of the simulation cell containing all 7x7 dots.”

Notice that previously we were referring to 4 dots (2x2) and 49 dots (7x7), because we were considering the grid architecture. We changed this to 2 dots and 7 dots in Fig. 1h as it makes the message clearer.

I did not have any idea that the authors were talking about the grid before. Could the authors answer my question now why the correlation length is 500 nm for 49 dots, 100 nm for 4 dots and 10 nm for 1 dot?

32) Lines 464-470: I agree that the size of the quantum dots have major impacts but do the authors find that the size of the quantum dots have major impacts in this manuscript? If so, where?

We agree with reviewers 1 and 2 that we did not provide enough data to support this conclusion.

We perform new atomistic simulations comparing the dots simulated in the first part of the paper (diameter 15 nm) with larger quantum dots (diameter 18 nm) (See Supplementary Fig 5). The results show that the most concerning part is the larger wavefunction variability and dispersion of the dot centres. Unexpectedly, we didn't observed much variation in valley splittings, g-factors, or valley phases.

We still preserve this conclusion but now we focus on the higher tolerance of the wavefunction of smaller dots to disorder.

Line 667-678: "Secondly, the size of the quantum dots can have a major impact on its performance. The fractal structure of the interface roughness can explain partially this effect, as smaller dots and pitches between dots restrict electrons to regions of the interface with smaller amplitude of the roughness. The average number of trapped charges in the oxide near the dot region also decreases significantly. This leads to higher quantum dot uniformity, and less statistical dispersion in the centres of the wavefunction(see Supplementary Fig. 6). "

In this case, the authors should remove the sentence "Secondly, the size of the quantum dots can have a major impact on its performance." from Conclusions.

-What do the open rectangles in Figure 3c signify? Please provide a detailed explanation of all the data, ensuring the legends are correctly assigned.

Reviewer #1 (Remarks on code availability):

I cannot find the code on <https://doi.org/10.6084/m9.figshare.23507439>

.

Reviewer #2 (Remarks to the Author):

The manuscript revision, initially strongly focused on the Si qubit properties dependence on the Si/SiO₂ interface roughness, is now extended to also include a discussion about the charge traps, inherently present at the Si/SiO₂ interface. The impact of these traps on the one-qubit and the two-qubit properties are convincingly discussed. Besides, the exchange coupling simulations, which now includes this property, are in agreement with the experimental data. I believe that this study can be of use to the wide community working on silicon qubits, from the atomistic comprehension of the qubit properties, to the design of electronic circuits that will eventually control the gate voltages. Overall, the manuscript has been greatly enhanced, and my initial concerns have been addressed. I recommend this work for publication in Nature Communications.

Reviewer #3 (Remarks to the Author):

The manuscript "Bounds to electron spin qubit variability for scalable CMOS architectures" has been notably improved. In its current state the manuscript is in my opinion publishable, but I think it might help to consider a few possible improvements that will be suggested in the following. I will skip over all the previously raised issues that the authors have addressed satisfactorily and focus on the issues I still consider relevant.

I will mark any comments from the rebuttal in >< before responding to them.

>We agree that this is the case. A more accurate statement is the following: in an idealised theoretical model where charge traps are the only issue and interface roughness is absent, charge traps would not impact qubit variability. Alternatively, one might state that the main impact of charge traps is to impact the electrostatic quantum dot formation, which impacts the qubit through the exposition of the spin to a different Si/SiO₂ interface environment. We have added an entire section about this in the new manuscript (and Fig. 4) talking about the role of charged traps on one-qubit properties. We also included charged traps in two-qubit simulations, which turned out into a better agreement with experimental data.<

The new Figure 4 is indeed very interesting. However, what puzzles me about these data is that based on Fig.4a qubits 8, 9, 13, 14, 41 and 42 have essentially no charges trapped in their direct vicinity. As I would have expected, the change in valley splitting and g factor for qubits 13, 14, 41 and 42 are very minor. However, for qubit 8 and 9 the valley splitting - in particular - changes dramatically. Do the authors have an explanation for this?

Regarding, Fig4c and 4d: I would also like to note that they are somewhat difficult to read right now. I would suggest considering a vertical guiding line for each or each 2nd qubit instead of just for each 10th as a potential improvement.

>The current yield per gate of our process in a University cleanroom is 95%.<

I would recommend to the authors to mention this somewhere (e.g. in the supplementary information) as it is a crucial detail that underlines the robustness of the presented work.

>We have now included the full distribution in the main text, on the side Fig 2f.<

There is something wrong with Figure 2 in the manuscript I got. Most notably, the caption refers to Fig. 2h but 2h that one does not seem to exist in the actual Figure (it only goes from a-g).

>CMOS architectures refer to metal-oxide-semiconductor, which is precisely what is studied here. What the reviewer considers “CMOS-compatible” is unclear from their comments. We believe we addressed the whole gamut of spin qubit quantum dot materials in our responses above.<

This is one of the gripes I still have with the manuscript which that the nomenclature is somewhat misleading.

Yes, the authors qubits are mostly based on a Pd/Ti (metals), AlO_x/SiO₂ (oxides), silicon (semiconductor) stack and hence it is fair to name them MOS qubits. Note, however, that for example the qubits in reference 3 are based on a Pd/Ti (metals), Al₂O₃ (oxide), SiGe/Si/SiGe (semiconductors) stack and hence are also technically MOS qubits. (The same thing can be said about a wide variety of other qubits.)

Furthermore, neither qubit should be called a CMOS qubit as such a thing – strictly speaking - shouldn't exist; in the same sense that a CMOS transistor doesn't exist. There are only nMOS and pMOS transistors that can be combined into a CMOS circuit. So technically the qubits presented in this manuscript should be called nMOS qubits. They are CMOS compatible – meaning that there is no known obstacle to monolithically integrating them with CMOS circuits (in a CMOS fab). The same thing is true for the Si/SiGe qubits though – nMOS qubits that are CMOS compatible (Ge/SiGe qubits would be pMOS qubits by that logic). Of course, the integration with CMOS circuits is notably easier for the Si/SiO₂ qubits than for the SiGe/Si/SiGe ones.

I understand that this is not how the nomenclature is used in the semiconductor qubit subfield and in the current manuscript it is mostly clear from context what the authors mean. I do think the nomenclature in the manuscript can be improved though particularly with a broad audience in mind.

>The Si/SiO₂ interface is at the heart of a multi-trillion-dollar industry and has shaped our era. Addressing how this interface affects spin qubits is of general interest to all scientists in our opinion.<

I am not sure what the authors are trying to accomplish here. This part of the review gave them a chance to justify why their work would be of interest for a broad community of scientists and they chose to answer with this.

Leaving aside that the “a bit more than half-a-trillion dollar” industry that works with the Si/SiO₂ interface does thus far not have a single notable commercial product that is related to the issues addressed by the manuscript: Concrete is a bigger industry than all of semiconductors combined and has shaped human history for significantly longer. Nevertheless, the vast majority of papers on concrete are published in very specialized journals. Not in spite of but exactly because concrete is such a topic of high importance that is followed by a very well-defined community.

The idea that all scientists care about spin-transport near the Si/SiO₂ interface is such a comical overstatement (think e.g. marine biologists or material scientists working on concrete) that I am not even sure whether this is meant to be taken seriously.

In my mind this is a manuscript that takes a deep dive into a very specific (and very important) issue of a specific qubit architecture using the most state-of-the-art simulation tools currently available to address that issue. However, while I fully agree that this is good work with crucial importance to the people working on this specific (and closely) related qubit architecture(s) - I struggle to see that the work presented is of notable interest for anyone not working on qubits.

Based on the comments in the review and my understanding the methods don't readily transfer to the study of other interfaces (e.g. the Si/SiGe interface) or other transport phenomena (e.g. electrical). The simulation methods used are state-of-the-art but not novel or so notably improved that other fields are likely to profit.

So, I still think that my original assessment that this work is better suited for a more specialized journal is fair.

Please see our answers for this new revision in blue

Reviewer's new comments are in black font.

Comments from previous revisions are included with an indent followed by *.

Reviewer 1

Reviewer #1 (Remarks on code availability):

1) I cannot find the code on <https://doi.org/10.6084/m9.figshare.23507439>

We have shared the data analysis code in the FigShare platform. We are not sure when will the editors make the code available, but they might be able to share with the reviewer beforehand if there is interest.

We note that, in accordance to our Code Availability statement, the code is going to be made available for any readers upon request. However, parts of the code are deemed confidential given their commercial sensitivity. All results can be reproduced by the community based on the materials included in this paper and in the references therein.

2) The authors have addressed some questions, but perhaps not in full depth. I believe that preparing the manuscript with greater attention to detail would enhance its appeal. Without this, readers may find it challenging to grasp the claims the authors are trying to make. Below, I have outlined my questions corresponding to specific points.

We thank the reviewer for their comments and apologize for not having addressed some of these issues thoroughly before.

3) In the manuscript, the authors mention a 49-qubit system, which naturally leads readers to wonder about the specific configuration of the gates for these qubits. It's crucial to consider that if the gate arrangement differs from what is described in your paper, it might not be straightforward to extrapolate your results directly to a 49-qubit system. Therefore, it would be beneficial to discuss or address this aspect somewhere in your work to clarify any potential discrepancies or assumptions regarding gate layouts for larger qubit arrays.>

While Fig 1a refers to an idealization of the 49 qubit array, we understand the reviewer's concern in regard to the feasibility of such gate array in practice, which will inform the reader about the realistic challenges of obtaining such dense array. We have now addressed this issue more thoroughly in the main text by adding the following sentences:

Lines 238-260: "In this paper, the quantum dots are simulated in a 7x7 grid array. Other architectures are also being explored^{~\cite{SpiderWeb2022}}, in part, because the practical design of such a dense array requires a sophisticated fabrication process with multiple metal layers to route the signals to the gates. While dense wiring in multiple metal layers is routinely integrated in front-end-of-line industrial processes, qubit demonstrations using this dense integration have only recently been explored

\cite{HRL_Sledge_2022}. Moreover, this dense array leaves no space for interspersed readout devices such as single-electron transistors, and would be dependent on a gate-based readout approach \cite{Gonzales_Readout_2021} or would require quantum information to be shuffled to the edges of the array for readout \cite{Peta_swap_sigillito_coherent_2019}. Our results for qubit variability are not drastically affected by the choice of a grid array, except for a small degree of nearest neighbor correlations (see Supplementary Fig.~6), which once simulated across the full 450 \times 450 nm Si/SiO₂ computer-generated interface, provides us with sufficient sampling to obtain statistical analyses accurately.

4) I did not have any idea that the authors were talking about the grid before. Could the authors answer my question now why the correlation length is 500 nm for 49 dots, 100 nm for 4 dots and 10 nm for 1 dot?

We apologize for this confusion. Indeed, in the first version of the paper we made Fig.1h-i considering a grid architecture with 49 dots and a rough 2D surface covering that region. The correlation lengths ($\lambda = \frac{2\pi}{q}$) where, however, estimated from the 1D power spectral density (PSD^{1D}(q)), which is the data that measure from the TEM's, with λ representing the wavelength of the oscillations in the system. To provide a sense of how λ is compared with the most relevant length scales, we marked these points Fig.1h (Dot width (10-15 nm), double dot length: (80-100 nm), lateral width of 7*7 architecture (500 nm)) and associated them with the correlation length of 1 dot, 4 dots and 49 dots.

After reviewer's comments, we realized that relating a 1D correlation length with the number of dots in a 2D architecture could be misleading, especially because of the assumption of a particular type of qubit array. We corrected this error in the last revision of the manuscript, in which the correlation length is only compared with 1D length-scales:

Caption of Fig. 1h: "This allows us to compare λ with most relevant length scales, namely the silicon lattice parameter silicon lattice parameter ($a_{\text{Si}} = 0.357 \text{ nm}$), the dot diameter (10-15 nm), the double dot length (80-100 nm) and the lateral length of the simulation cell containing all 7x7 dots (500 nm)"

5)

* Fig. 3b: What are the open triangles? Where are black rectangles and open rectangles used?

*We have modified the legend of Fig. 3b for clarity. All filled markers are data from the qubit 1 of each device. Empty markers are data from qubit two. The device is identified by the marker in Fig. 3a.

I still find it challenging to interpret the legends. Could you explain what the open triangles represent? Also, what do the open rectangles in Figure 3c signify? Please provide a detailed explanation of all the data, ensuring the legends are correctly assigned.

We included a detailed explanation in the caption of Fig 3a-b

Caption of Fig 3a-b: "In **a** we compare the difference between the Larmor frequencies of the two-qubit qubits measured in each device $(g_1 - g_2)\mu_B$ vs the differences between the frequency of neighbouring dots simulated atomistically (Fig. **1b**). The marker for experimental data is associated with the device (A to E). In **b**, we compare the top gate Stark shift $d\epsilon/dV$ measured in the two qubits of each device, with atomistic simulations of the dots (methods). Two qubits in a device have a different Stark shift $d\epsilon/dV$ due to variations in surface roughness. Filled markers represent the data for the

first qubit and empty markers for the second qubit (e.g. the empty purple triangles represents the Stark shifts measure on the second qubit of device E.)”

The purple open triangle would be the data for the second qubit of device E, while filled triangles represent the data for the first qubit in the same device.

6)

*Lines 464-470: I agree that the size of the quantum dots have major impacts but do the authors find that the size of the quantum dots have major impacts in this manuscript? If so, where?

*We agree with reviewers 1 and 2 that we did not provide enough data to support this conclusion.

We perform new atomistic simulations comparing the dots simulated in the first part of the paper (diameter 15 nm) with larger quantum dots (diameter 18 nm) (See Supplementary Fig 5). The results show that the most concerning part is the larger wavefunction variability and dispersion of the dot centres. Unexpectedly, we didn't observed much variation in valley splittings, g-factors, or valley phases.

We still preserve this conclusion but now we focus on the higher tolerance of the wavefunction of smaller dots to disorder.

Line 667-678: “Secondly, the size of the quantum dots can have a major impact on its performance. The fractal structure of the interface roughness can explain partially this effect, as smaller dots and pitches between dots restrict electrons to regions of the interface with smaller amplitude of the roughness. The average number of trapped charges in the oxide near the dot region also decreases significantly. This leads to higher quantum dot uniformity, and less statistical dispersion in the centres of the wavefunction (see Supplementary Fig. 6). “

In this case, the authors should remove the sentence “Secondly, the size of the quantum dots can have a major impact on its performance.” from Conclusions.

We agree with reviewers' comment and remove the conclusion about dot size from the manuscript.

Reviewer #2 (Remarks to the Author):

The manuscript revision, initially strongly focused on the Si qubit properties dependence on the Si/SiO₂ interface roughness, is now extended to also include a discussion about the charge traps, inherently present at the Si/SiO₂ interface. The impact of these traps on the one-qubit and the two-qubit properties are convincingly discussed. Besides, the exchange coupling simulations, which now includes this property, are in agreement with the experimental data. I believe that this study can be of use to the wide community working on silicon qubits, from the atomistic comprehension of the qubit properties, to the design of electronic circuits that will eventually control the gate voltages. Overall, the manuscript has been greatly enhanced, and my initial concerns have been addressed. I recommend this work for publication in Nature Communications.

We thank the reviewer for the valuable contributions that led to a very significant improvement of this paper. We are glad that the reviewer recommended this article for publication.

Reviewer #3 (Remarks to the Author):

1) The manuscript "Bounds to electron spin qubit variability for scalable CMOS architectures" has been notably improved. In its current state the manuscript is in my opinion publishable, but I think it might help to consider a few possible improvements that will be suggested in the following. I will skip over all the previously raised issues that the authors have addressed satisfactorily and focus on the issues I still consider relevant.

We are glad that the reviewer considers that our paper has improved significantly, and that the manuscript is publishable.

2) I will mark any comments from the rebuttal in >< before responding to them.

*>We agree that this is the case. A more accurate statement is the following: in an idealised theoretical model where charge traps are the only issue and interface roughness is absent, charge traps would not impact qubit variability. Alternatively, one might state that the main impact of charge traps is to impact the electrostatic quantum dot formation, which impacts the qubit through the exposition of the spin to a different Si/SiO₂ interface environment. We have added an entire section about this in the new manuscript (and Fig. 4) talking about the role of charged traps on one-qubit properties. We also included charged traps in two-qubit simulations, which turned out into a better agreement with experimental data.<

3) The new Figure 4 is indeed very interesting. However, what puzzles me about these data is that based on Fig.4a qubits 8, 9, 13, 14, 41 and 42 have essentially no charges trapped in their direct vicinity. As I would have expected, the change in valley splitting and g factor for qubits 13, 14, 41 and 42 are very minor. However, for qubit 8 and 9 the valley splitting - in particular - changes dramatically. Do the authors have an explanation for this?

This question is very interesting. We investigated this matter further and included a plot in the supplementary. While it is true that the simulated dots number 8 and 9 are quite far from any charge trap in the simulation, we conclude that the significant change in the valley splitting is due to the susceptibility of this parameter to electric fluctuations, which is higher when the valley splitting of the original dot is already large.

To show this, we plotted in the Figure below (included as Fig. 7 in the supplementary) the distribution of valley splittings VS vs its difference before and after the inclusion of charged traps ($VS_{\text{Traps}} - VS$). We notice that a large VS is very correlated with a large ($VS_{\text{Traps}} - VS$), which is expected considering the non-linear dependence of the valley splitting on the electric field which was already explored in Fig 2f. In turn, the valley splitting of a dot with large VS is typically more susceptible to the change of electrostatic configuration introduced by the presence of charged traps.

Figure 1: Copy of the new Fig 7 from the supplementary material but highlighting specifically dots 8 and 9 in response to reviewer’s comments. The data shows the distribution of valley splittings (VS) for the 49 dots simulated compared with the difference in valley splitting before and after the inclusion of negative traps in the simulation ($VS_{\text{Traps}} - VS$).

Please note that this high electric susceptibility of the valley splitting is usually innocuous for qubit experiments, as the qubit is encoded in the spin and the only requirement for the valley state is that it is higher than the Zeeman splitting. The spin response to electric fields is also very different as described in Fig. 3 of the paper. The g-factor has a very small variation in dots 8 and 9, for instance (Fig. 4c).

We added the following text to the main document citing Supplementary Fig. 7:

Lines 554-558: “This increase is typically larger for quantum dots that already had a large valley splitting (See Supplementary Fig. 7) due to their higher susceptibility to electric fluctuations. “

4) Regarding Fig4c and 4d: I would also like to note that they are somewhat difficult to read right now. I would suggest considering a vertical guiding line for each or each 2nd qubit instead of just for each 10th as a potential improvement.

We modified Fig4c-d as indicated by the reviewer.

5)

*>The current yield per gate of our process in a University cleanroom is 95%.<

I would recommend to the authors to mention this somewhere (e.g. in the supplementary information) as it is a crucial detail that underlines the robustness of the presented work.

We thank the referee for the comment. We do not have sufficient statistics to back up the claim of this yield – it is based on a rough estimate of the measurement campaigns we have done lately. To allude to the general qualitative role that the improved yield has in our ability to reach these conclusions, we have changes the first part of the Methods section that used to be called “Oxide growth.”

Lines 789-797: “**Fabrication process.** The SiO₂ gate oxide (7.5-8.0 nm) was thermally grown on the silicon surface in a custom-built high-quality oxide furnace as part of a standard MOS device fabrication process. The gate fabrication process was iterated multiple times to improve yield, which was an enabling feature

for this study, leading to the successful formation of several devices with nominally identical layouts.”

6)

*>We have now included the full distribution in the main text, on the side Fig 2f.<

There is something wrong with Figure 2 in the manuscript I got. Most notably, the caption refers to Fig. 2h but 2h that one does not seem to exist in the actual Figure (it only goes from a-g).

We corrected the bug in the captions of Fig. 2.

7)

*>CMOS architectures refer to metal-oxide-semiconductor, which is precisely what is studied here. What the reviewer considers “CMOS-compatible” is unclear from their comments. We believe we addressed the whole gamut of spin qubit quantum dot materials in our responses above.<

This is one of the gripes I still have with the manuscript which that the nomenclature is somewhat misleading.

Yes, the authors qubits are mostly based on a Pd/Ti (metals), AlO_x/SiO₂ (oxides), silicon (semiconductor) stack and hence it is fair to name them MOS qubits. Note, however, that for example the qubits in reference 3 are based on a Pd/Ti (metals), Al₂O₃ (oxide), SiGe/Si/SiGe (semiconductors) stack and hence are also technically MOS qubits. (The same thing can be said about a wide variety of other qubits.) Furthermore, neither qubit should be called a CMOS qubit as such a thing – strictly speaking - shouldn’t exist; in the same sense that a CMOS transistor doesn’t exist. There are only nMOS and pMOS transistors that can be combined into a CMOS circuit. So technically the qubits presented in this manuscript should be called nMOS qubits. They are CMOS compatible – meaning that there is no known obstacle to monolithically integrating them with CMOS circuits (in a CMOS fab). The same thing is true for the Si/SiGe qubits though – nMOS qubits that are CMOS compatible (Ge/SiGe qubits would be pMOS qubits by that logic). Of course, the integration with CMOS circuits is notably easier for the Si/SiO₂ qubits than for the SiGe/Si/SiGe ones.

I understand that this is not how the nomenclature is used in the semiconductor qubit subfield and in the current manuscript it is mostly clear from context what the authors mean. I do think the nomenclature in the manuscript can be improved though particularly with a broad audience in mind.

We appreciate the reviewer’s position and agree that the broad readership of Nature Communications may be confused by this use of jargon. Instead of using “CMOS compatibility”, we will therefore explicitly discuss the context of interest as “qubits formed in traditional silicon/silicon dioxide interface, compatible with the high-yield integration of on-chip electronic components”.

We include this information in the first paragraph.

Lines 12-30: “In particular, the similarity between quantum dots defined by gate electrodes on top of a silicon/silicon dioxide interface and the MOSFET transistors in materials, design, and fabrication enables the integration of manufacturing techniques exclusive to semiconductor foundries onto the scaling of quantum processors.

Here, we specifically treat the case such qubits, formed in quantum dots at the Si/SiO₂ interface, which

are compatible with the high-yield integration of on-chip electronic components, and refer to these as CMOS spin qubits. We note, however, that other forms of silicon-based quantum dots can be manufactured, for instance by leveraging a Si/SiGe quantum well^{~\cite{philips_universal_2022}}. These materials present their own complex challenges and advantages and impose significantly different architectural choices compared to the Si/SiO₂ interface, and hence are left out of this investigation.”

8)

*>The Si/SiO₂ interface is at the heart of a multi-trillion-dollar industry and has shaped our era. Addressing how this interface affects spin qubits is of general interest to all scientists in our opinion.<

I am not sure what the authors are trying to accomplish here. This part of the review gave them a chance to justify why their work would be of interest for a broad community of scientists and they chose to answer with this.

Leaving aside that the “a bit more than half-a-trillion dollar” industry that works with the Si/SiO₂ interface does thus far not have a single notable commercial product that is related to the issues addressed by the manuscript: Concrete is a bigger industry than all of semiconductors combined and has shaped human history for significantly longer. Nevertheless, the vast majority of papers on concrete are published in very specialized journals. Not in spite of but exactly because concrete is such a topic of high importance that is followed by a very well-defined community.

The industry we refer to is the CMOS industry, not specifically quantum. We understand the reviewers point, but respectfully disagree with the comparison between CMOS and concrete. We leave it to the editors to decide whether they consider this technological platform a point of interest for their journal.

9) The idea that all scientists care about spin-transport near the Si/SiO₂ interface is such a comical overstatement (think e.g. marine biologists or material scientists working on concrete) that I am not even sure whether this is meant to be taken seriously.

In my mind this is a manuscript that takes a deep dive into a very specific (and very important) issue of a specific qubit architecture using the most state-of-the-art simulation tools currently available to address that issue. However, while I fully agree that this is good work with crucial importance to the people working on this specific (and closely) related qubit architecture(s) - I struggle to see that the work presented is of notable interest for anyone not working on qubits.

We do not have anything to add to this point. We thank the reviewer for their appreciation of the importance of our work and leave it to the editors of Nature Communications the judgement about whether scientists in other fields are interested or not in the research being undertaken in the field of quantum computing.

10) Based on the comments in the review and my understanding the methods don't readily transfer to the study of other interfaces (e.g. the Si/SiGe interface) or other transport phenomena (e.g. electrical). The simulation methods used are state-of-the-art but not novel or so notably improved that other fields are likely to profit. So, I still think that my original assessment that this work is better suited for a more specialized journal is fair.

We believe the reviewer is mistaken about our research's content. We did not discuss transport phenomena in the current work.

We have now addressed all the excellent technical comments from the reviewer, including an extensive simulation campaign that indeed led to valuable new insight. We thank the reviewer for all their input, which led to a much improved manuscript. We respectfully disagree with the reviewer's view on the importance of our research and its appeal to other scientists. It is our view that the level of broad knowledge and scientific curiosity of the readership of Nature Communications is better addressed by their editors and leave it to them to form their judgement, making ourselves available for any further clarifications.

REVIEWERS' COMMENTS

Reviewer #1 (Remarks to the Author):

All my questions have been addressed. I recommend this article for publication.